

# Amplified surface warming in the Southwest Pacific during the mid-Pliocene (3.3–3.0 Ma) and future implications

Georgia R. Grant[1], Jonny H.T. Williams[2], Sebastian Naeher[1], Osamu Seki[3], Erin L. McClymont[4], Molly O. Patterson[5], Alan M. Haywood[6], Erik Behrens[2], Masanobu Yamamoto[3], Katelyn Johnson[1]

[1]GNS Science, Lower Hutt, New Zealand
[2]NIWA, Wellington, New Zealand
[3]Hokkaido University, Sapporo, Hokkaido, Japan
[4]Durham University, Durham, United Kingdom
[5]Binghamton University, SUNY, New York, USA
[6]Univerisity of Leeds, Leeds, United Kingdom

*Correspondence to*: Georgia R. Grant (G.Grant@gns.cri.nz)

**Abstract**

Based on Nationally-Determined Contributions concurrent with Shared Socio-economic Pathway (SSP) 2-4.5, the IPCC predicts global warming between 2.1–3.5°C (very likely range 10[th]-90[th] percentile) by 2100 AD. However, global average temperature is a poor indicator of regional warming and Global Climate Models (GCMs) require validation with instrumental or proxy data from geological archives to assess their ability to simulate regional ocean and atmospheric circulation, and thus, to evaluate their performance for regional climate projections. The Southwest Pacific is a region that performs poorly when GCMs are evaluated against instrumental observations. The New Zealand Earth System Model (NZESM) was developed from the United Kingdom Earth System Model (UKESM) to better understand Southwest Pacific response to global change, by including a nested ocean grid in the Southwest Pacific with 80% greater horizontal resolution than the global-scale host.

Here, we reconstruct regional Southwest Pacific sea surface temperature (SST) for the mid-Pliocene Warm Period (mPWP; 3.3–3.0 Ma), which has been widely considered a past analogue with an equilibrium surface temperature response of +3°C to an atmospheric $CO_2$ concentration of ~350–400ppm, to assess the warming distribution in the Southwest Pacific. This study presents proxy SSTs from seven deep sea sediment cores distributed across the Southwest Pacific. Our reconstructed SSTs are derived from molecular biomarkers preserved in the sediment - alkenones (i.e., $U_{37}^{K'}$ index) and isoprenoid glycerol dialkyl glycerol tetraethers (i.e., $TEX_{86}$ index) and are compared with SSTs reconstructed from the Last Interglacial (125 ka), Pliocene Model Intercomparison Project (PlioMIP) outputs and transient climate model projections (NZESM and UKESM) of low to high range SSPs for 2090-2099 AD.

Mean interglacial equilibrium SSTs during the mPWP for the Southwest pacific sites, were on average, 4.2°C (1.8–6.1°C likely range) above pre-industrial and show good agreement with model outputs from NZESM and UKESM under mid-range SSP 2-4.6 conditions. These results highlight that not only is the mPWP an appropriate analogue when considering future temperature change in the centuries to come, but also demonstrate that the Southwest Pacific region will experience warming that exceeds that of the global mean if atmospheric $CO_2$ remains above 350 ppm.



## 1 Introduction

The latest IPCC climate projections to 2100 AD project average global surface warming of between 1.4-4.4°C depending on the emissions pathway (IPCC, 2022). While limiting global warming to 1.5°C urgently requires policies and actions to bring about steep emission reductions this decade, global warming could be stabilised at

2.0°C, if the latest Nationally Determined Contributions are achieved (Meinshausen, 2022). Despite stabilising at 2.0°C, heat taken up by the ocean and the polar ice sheets would ensure global sea-level would continue to rise for centuries to come (IPCC, 2022). Warming above 2.0°C may trigger rapid unstoppable collapse of the marine-based sectors of the Antarctic Ice Sheets, with one model for a high-emissions scenario suggesting global mean sea-level rise of up to 2 m by 2100 AD and 13 m by 2300 AD (DeConto *et al*., 2021; IPCC, 2022).

Notwithstanding the high-end scenarios, a stability threshold for Antarctic ice shelves is crossed above +2.0°C that commits the planet to multi-metre, multi-century sea-level rise (DeConto and Pollard, 2016; Golledge *et al*., 2019; Lowry *et al*., 2021). Additionally, the regional expression of global warming can differ significantly from global averages, as is evident from most land regions currently recording warming which exceeds the global average (Hoegh-Guldberg *et al*., 2018; Sutton and Bowen, 2018; Doblas-Reyes *et al*., 2021). Regionally focussed

climate models are necessary for island nations with oceanic influence and dramatic topography such as New Zealand, since these parameters are unresolvable at the spatial resolutions used by climate models with a low, uniform resolution (Doblas-Reyes *et al*., 2021).

Here, we consider the regional climate of the Southwest Pacific and Southern Ocean, which is often

misrepresented due to coarse resolution and biases introduced in global climate models (Behrens *et al*., 2020, 2022; Williams *et al*., 2021). Steep regional gradients in SST, salinity and nutrients, characterise water masses spanning the Southwest Pacific and New Zealand continent (Zealandia - Te Riu-a-Māui) (Ridgway, 2007; Chiswell *et al*., 2015; Chiswell, 2021), which represents a key location for southward heat transport balanced by northward flow of deep western boundary currents (Carter *et al*., 2004). Subtropical waters are transported

southward through surface eddies and the East Australian Current and Tasman Front (Fig 1; Behrens *et al.,* 2019). Zealandia is situated at the confluence of relatively cool, fresh, nutrient-rich Subantarctic Waters and warm, salty, nutrient-poor Subtropical Waters, defining the Subtropical Front (e.g., Chiswell *et al*., 2015; Fig. 1). The NZESM was developed from its parent model, the UKESM, to address the need for higher spatial resolution in models across Zealandia (Williams *et al*., 2016). An increased horizontal grid resolution from 1° to 0.2° better simulates

boundary currents and surface eddies, and result in an increased meridional heat transport from the equator to higher southern latitudes (Behrens *et al*., 2019) and is in better agreement with historical observations compared to the UKESM (Behrens *et al*., 2020).

Past climate data allow the reconstruction of the equilibrium climate states in response to both fast and slow Earth

system feedbacks involving the cryosphere, ocean and atmospheric circulation and the carbon cycle. Data from these geological archives for times representing higher-than-present $CO_2$ worlds have been widely used in climate model-intercomparison projects (CMIPs) to assess the performance of transient GCMs run to equilibrium (e.g. Haywood *et al*., 2019; Masson-Delmotte *et al*., 2013). While most CMIPs reconcile global mean temperatures, they poorly reconcile regional climatic patterns such as polar amplification (Naish & Zwartz, 2012; Haywood *et*

*al*., 2019; Masson-Delmotte *et al*., 2013; Fischer *et al*., 2018). This is in part due to the incomplete spatial coverage





of the geological data, accuracy and quality of the data, the resolution of GCM grids and their treatment of mid- to high-latitude polar processes. Equilibrium Climate Sensitivity (ECS) (model warming associated with a doubling of $CO_2$ once the energy balance has reached equilibrium) is one important measure of how models perform on longer timescales. An increase in ECS from CMIP Phase 5 to CMIP Phase 6 ensemble has been linked

to shortwave cloud feedbacks, which has significant impact over the Southern Ocean (Zelinka *et al*., 2020; Zhu *et al*., 2021). Higher ECS is more consistent with estimates of paleo climate sensitivity (Kageyama *et al.,* 2018). We assess the magnitude and distribution of warming for the Southwest Pacific for various emissions scenarios and discuss the differences between the global climate models and paleoclimate reconstructions and consider the implications for interpreting projections of future warming in the SW Pacific.


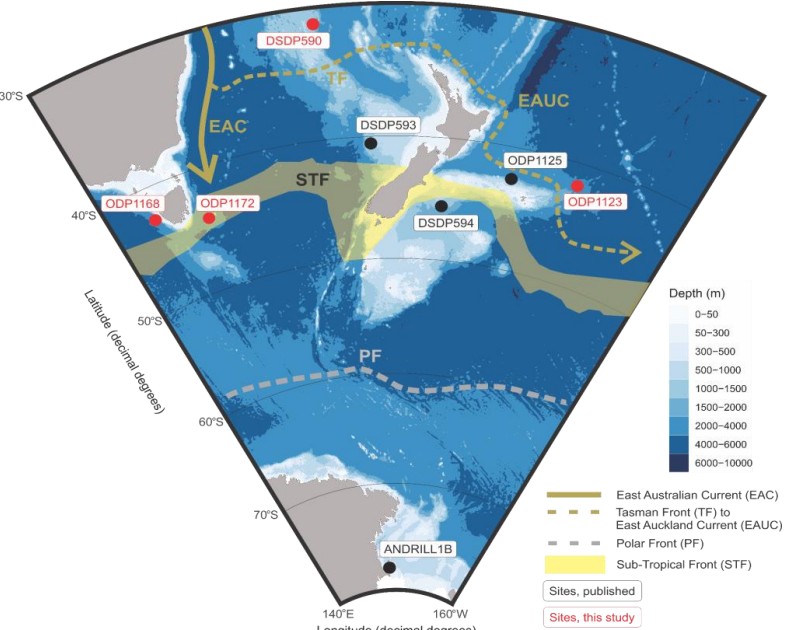

**Figure 1: Location map for sites used in the Southwest Pacific Sea Surface Temperature (SST) reconstruction (North**
**top of page). Sites in black have been previously published and sites in red were analysed in this study. Present day surface ocean circulation and fronts referenced in text are displayed. Note ODP 806 (0.3° N 159.4° E) is not displayed in this projection. Bathymetry is plotted using 'ggOceanMaps' (Vihtakari, 2022) and bathymetry data are sourced from Amante and Eakins (2009).**


## 1.1 Paleoclimate analogues

*Mid-Pliocene Warm Period (3.3 – 3.0 Ma)*

Climatic conditions last experienced during the mPWP (3.3–3.0 Ma) may be reached by 2100 AD if emissions are abated in line with the SSP2-4.5 scenario, which is the pathway aligned to current policy (not the aspirational



1.5°C Paris-target) (Burke *et al*., 2018). The mPWP spans a 300 kyr period when atmospheric $CO_2$ was comparable to present day (mean 390 ppm; Chalk *et al*., 2017; De La Vega *et al*., 2020), During this period interglacial global temperatures were 2–3°C warmer (Dowsett *et al*., 2013; Masson-Delmotte *et al*., 2013), and the amplitude of glacial-interglacial sea-level change was likely between 6 and 17 m (16th-84th percentile) (Grant *et al*., 2019; Grant and Naish, 2021). Such a rise in global sea-level implies melting of the Greenland Ice Sheet

(Koenig *et al*., 2015; Batchelor *et al*., 2019), West Antarctic Ice Sheet (Naish *et al.,* 2009; McKay *et al*., 2012) and parts of marine-based East Antarctic Ice Sheet (Cook *et al*., 2013; Patterson *et al.,* 2014; Bertram *et al*., 2018). Therefore, the interglacial periods of mPWP are considered to be the most accessible and suitable past analogue, or window, into the future equilibrium response of the Earth system to warming in line with SSP2-4.5 (Naish & Zwartz, 2012; Dowsett *et al*., 2013; Haywood *et al*., 2019).


The mPWP has been the focus of several major international research initiatives. The Pliocene Research, Interpretation and Synoptic Mapping (PRISM) project (Dowsett *et al*., 2013; 2016) undertook a global compilation of paleoclimate data, primarily surface temperature reconstructions. The Pliocene Modelling Intercomparison Project (PlioMIP; Haywood *et al*., 2016) made comparisons between PRISM data (average

interglacial temperatures over the 300 ky-duration period) and a suite of climate models, finding ECS to be 2–3°C (Haywood *et al*., 2012; Masson-Delmotte *et al*., 2013). Subsequently, Marine Isotope Stage (MIS) KM5c (3.2 Ma) interglacial became a focus for reconstructing warming within mPWP as insolation values and the orbital configuration were most similar to the Holocene interglacial (Haywood *et al*., 2020; McClymont *et al*., 2020). While, based on less data points, this approach revealed a higher ECS of 2.6–4.8°C for conditions of MIS KM5c

from the PlioMIP Phase 2 ensemble (PlioMIP2; Haywood *et al*., 2020). A recent review of SSTs in the mPWP for MIS KM5c by the PlioVAR working group (Pliocene climate variability on glacial-interglacial timescales; McClymont *et al*., 2020) used alkenones to reconstruct an average global SST warming of 3.2–3.4°C above pre-industrial SST. This is slightly warmer than PlioMIP2 simulations, where global surface air temperature over oceans were ~2.8°C above pre-industrial. However, differences are suggested to be due to regional ocean

circulation and proxy signals (McClymont *et al*., 2020).

While interglacial minima and glacial maxima in the benthic $\delta^{18}O$ stack (MISs) have been the primary means of reconstructing the timing and magnitude of global sea-level variations over the last 5 Ma (Lisiecki and Raymo, 2005), for some time intervals (i.e., mPWP) global sea-level is known to fluctuate at a higher frequency than can

be assessed in the benthic $\delta^{18}O$ stack (Grant *et al*., 2019). This is also the case for other proxies with variable sampling resolution such as SST that have not been tuned to the $\delta^{18}O$ stack (e.g. Herbert *et al*., 2010; Fig. 2). The reliance on orbitally tuned timescales in deep ocean paleoclimate records has potentially led to the misinterpretations of the timing, frequency and amplitude of glacial-interglacial climate change. This is particularly the case in the Pliocene and Early Pleistocene where there are less globally distributed $\delta^{18}O$ records

and many are of coarse sampling resolution (Lisiecki and Raymo, 2005). In a number of studies (Lisiecki and Raymo, 2005; Miller *et al*., 2012; Grant *et al*., 2019), average glacial climate conditions (global surface temperature and sea-level) during the mPWP have been considered similar to those of the Holocene.



*Last interglacial (125 ka)*

Finally, we  briefly compare these results to the Last Interglacial MIS 5e (~125 ka) as many of the sites investigated here were also used by Cortese *et al.* (2013) in a proxy SST study. Peak interglacial SSTs were reconstructed from core-top planktonic foraminiferal assemblages, calibrated to modern SSTs and then applied to paleo assemblages (Cortese *et al.*, 2013). The Southwest Pacific study presented warming focused in Tasmania

and western New Zealand and proposed a strengthened East Australian Current bathing Tasmania with warmer water (Cortese *et al.*, 2013). MIS 5e represents a lower global average temperature increase of 1–2°C above pre-industrial in response to changing orbital configurations on radiative forcing (rather than $CO_2$), associated with 6-9 m of sea-level rise, which together with the mPWP analogue discussed above implies extreme sensitivity of the polar ice sheets  to relatively small changes in global mean surface temperature (Dutton *et al.*, 2013).


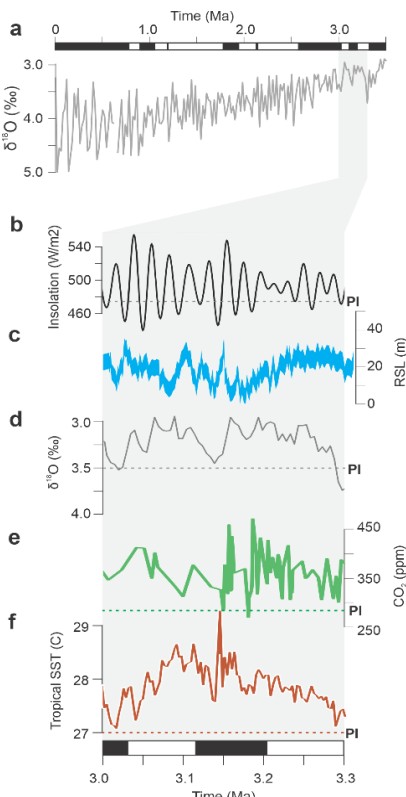

**Figure 2: mid-Pliocene Warm Period (mPWP) climate context showing reconstructions  of a) a combined signal of global sea-level and ocean temperature from deep sea benthic δ¹⁸O data (Lisiecki and Raymo, 2005) spanning mPWP**
**to present, b)  daily insolation at 65° N (21st June: Laskar *et al.*, 2004), c) global relative sea-level change from the PlioSeaNZ record, Whanganui Basin, New Zealand(Grant *et al.*, 2019), d) a combined signal of global sea-level and ocean temperature from deep sea benthic δ¹⁸O data mPWP (3.3-3.0 Ma) of (Lisiecki and Raymo, 2005), e) atmospheric $CO_2$ from δ¹¹B-pH proxy (De La Vega *et al.*, 2020) and f) tropical Sea Surface Temperatures (SSTs) from alkenone paleothermometry (Herbert *et al.*, 2010). Pre-industrial (PI) estimates are also shown.**




## 1.2 Future scenarios

Here we display model results of future projections from NZESM and UKESM. These are previously published and thus introduced here, while the comparisons to data presented in this study are discussed in detail in Section 3.

The NZESM (Williams *et al*., 2016, Behrens *et al*., 2020) is based on the UKESM (Sellar *et al*., 2019; Senior *et al*., 2020), a CMIP6 earth system model (ESM) containing a dynamic atmosphere, ocean, prognostic sea ice, complex atmospheric chemistry and ocean biogeochemistry. Via a two-way nesting scheme, ocean physical parameters were dynamically downscaled from 1° to 0.2° in the NZESM to better simulate boundary currents and mesoscale variability, instrumental for southward heat transport (Behrens *et al*., 2019). This nesting improves the

steady state simulated sea surface properties (Behrens *et al*., 2020; 2022). With the exception of a solar-cycle-dependence of the ozone photolysis scheme included in the NZESM (Dennison *et al*., 2019), the atmospheric physics is identical to the UKESM in all other respects. Globally averaged SSTs are marginally warmer than the UKESM in all pathways up to 2100 AD, but that difference is reduced as the magnitude of warming increases under higher-emission scenarios (Fig. 3). Indeed, for 2090–2099 AD in SSP3-7.0, the mean difference between

the two models is essentially zero for higher greenhouse gas levels. This global signal is, dominated by the southern hemisphere warming induced by increased southward heat transport from the tropics in the NZESM.

The latest climate projections are grouped according to primary Shared Socioeconomic Pathways (SSPs; Lee *et al*., 2021) forced by various greenhouse gas emissions and other radiative forcings and simulated by the CMIP6

(Eyring *et al*., 2016). These pathways are differentiated by degrees of very likely warming by 2100 AD, i.e. 1.3–2.4°C (SSP1 - sustainability), 2.1–3.5°C (SSP2 - middle of the road), 2.5–4.6°C (SSP3 - regional rivalry) (Chen *et al*., 2021; O'Neill *et al*., 2016). NZESM and UKESM were both run for SSP1-2.6, SSP2-4.5 and SSP3-7.0 and broadly correspond to low, medium and high emissions scenarios and were run out to 2100 AD. While UKESM was run for other SSPs, NZESM was not, so we have restricted comparison to these scenarios.


The UKESM and NZESM have an ECS of 5.4°C (Sellar *et al*., 2019) and is higher than the consensus assessment based on models and data which places ECS in the likely range (high confidence) of 2.5–4°C (Zelinka *et al*., 2020; IPCC, 2022). This is most clearly seen in the degree of global warming (Fig. 3a) compared to the regional warming of the high-resolution NZESM ocean-grid area (Fig. 3b). Climate scenarios by the 2090-2099 AD period

generated warming of i) ~3°C globally and ~2°C regionally for SSP1-2.6, ii) 4°C globally and 3°C regionally for SSP2-4.5, and iii) 6°C globally and 4.5°C regionally for SSP3-7.0 (Fig. 3). This differs (close to half) from mean global warming from the CMIP6 model ensemble of ~1.8°C, 2.7°C and 3.6°C for SSP1-2.6, 2-4.5 and 3-7.0 respectively by 2100 AD (IPCC, 2022). Annual mean SSTs were extracted for all sites and are reported here. Sites in the tropics (ODP 806) and Southern Ocean (ANDRILL) were excluded as they are outside of the NZESM

high-resolution ocean-grid region.

As a reference or pre-industrial control, the results generated from the Hadley Centre Global Sea Ice and Sea Surface Temperature (HadISST) model were used from 1870–1879 AD (NCAR, 2022; Rayner *et al*., 2003). This was selected as the most complete reanalysis product nearest to pre-industrial conditions and reduces inherited

bias from control model runs if we were to use UKESM pre-industrial control.




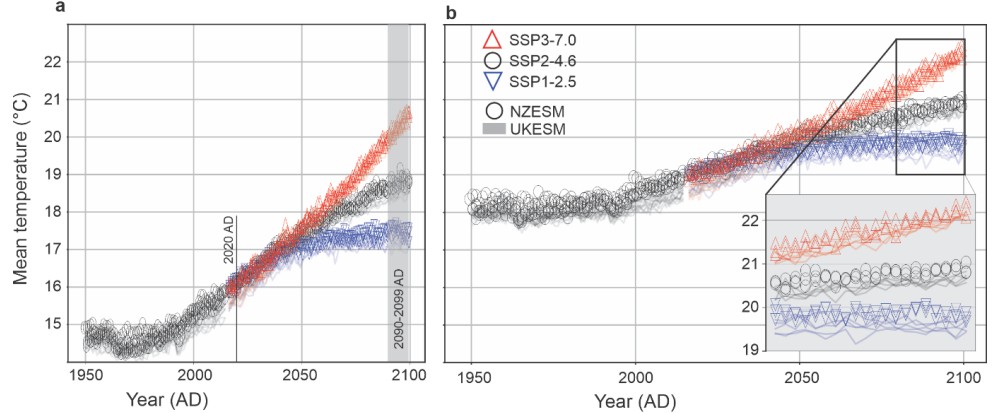

**Figure 3: Mean Surface Air Temperature from NZESM and UKESM simulations of low- to high-range emission Shared Socio-economic Pathways (SSPs) for a) the global region and b) the area covered by the high-resolution ocean grid of NZESM. Results generated from the UKESM (Sellar *et al.*, 2019) and NZESM (Williams *et al.*, 2016, Behrens *et al.*, 2022) projections used in this study are extracted for all SSPs for 2090–2099 AD.**

## 2 Methods

To enable comparison of past SSTs with future projections, we assess the full duration and glacial to interglacial amplitude of the mPWP, for sites across the Southwest Pacific region. In this approach, glacial and interglacial modal means are determined statistically to ensure the pattern and magnitude of warming is more representative of mPWP interglacial climate conditions as opposed to the single peak interglacial conditions that have been the focus of previous climate reconstructions (Dowsett *et al.*, 2016, Haywood *et al.*, 2020, McClymont *et al.*, 2020). We have applied the $U^{K'}_{37}$ index (unsaturated ketone index; Prahl and Wakeham, 1987) to reconstruct SSTs at four new sites (DSDP 590, ODP 1123, ODP 1168 and ODP 1172) which complement three sites with previously published $U^{K'}_{37}$-derived SSTs (DSDP 593: McClymont *et al.*, 2016; DSDP 594 and ODP 1125: Caballero-Gill *et al.*, 2019). We have also applied TEX$_{86}$ (TetraEther index of tetraether consisting of 86 carbon atoms) most commonly correlated with SST or shallow subsurface (50–200 m) temperatures (Tierney and Tingley, 2015) at two of the sites (DSDP 590 and ODP 1172). Two additional sites, ODP 806 (Eastern Equatorial Pacific: Medina-Elizalde and Lea, 2010) and ANDRILL (Ross Sea, Antarctica; McKay *et al.*, 2012) are located outside of the Southwest Pacific and provide a meridional climate context.

We extract site-specific simulated SSTs from PlioMIP and future UKESM and NZESM to compare the reconstructed pattern of warming in the Southwest Pacific during the mPWP.

### 2.1 mid-Pliocene Warm Period records

Sea surface temperature records from nine sites are presented in this study, including published SST data from five sites and new SST data from four sites to improve the geographical resolution across the Southwest Pacific and surrounding water masses. Inclusion of tropical site ODP 806 and Antarctic ANDRILL site in the Ross Sea



allows us to present a latitudinal transect from 0.3° N to 77° S, within longitudes 155° E to 165° W (Fig. 1; Table
1). Sites were selected from cores that were available through International Ocean Drilling Program (IODP) and
predecessor drilling programmes. Sampling of new sites was evenly distributed across the mPWP (Table S1 and
S2), with age models selected from the most up to date publications (Table 1). The age models used in previously
published SST records are retained here (Table 1). Published age models by Karas *et al*. (2011), Patterson *et al*.
(2016) and McClymont *et al*. (2016) are calibrated to the deep sea $\delta^{18}$O benthic stack (Lisiecki and Raymo, 2005).
In the case of sites ODP 1172 and ODP 1168, we use the integrated shipboard age models for the mPWP (Exon
*et al*., 2001). Linear interpolation of magnetostratigraphy provided by Exon *et al*. (2001) was used in absence of
high-resolution $\delta^{18}$O records for site ODP 1168 and 1172 that could be correlated to the deep sea $\delta^{18}$O benthic
stack.

Sediment samples obtained from four sites (ODP 1168, ODP 1172, ODP 1123, DSDP 590) were analysed for
alkenone-based SST reconstructions using the $U_{37}^{K'}$ index (e.g., Prahl and Wakeman, 1987; Section 2.2) at a target
temporal resolution of less than 10 kyr (Table 1). For two sites (DSDP 590 and ODP 1172), additional analysis
of glycerol dialkyl glycerol tetraethers (GDGTs) were undertaken to derive for TEX$_{86}$-based SST estimates (e.g.,
Schouten *et al*., 2002). $U_{37}^{K'}$ derived SSTs were reported previously for DSDP 594 and ODP 1125 (Caballero-Gill
*et al*., 2019; McClymont *et al*., 2020), and DSDP593 McClymont *et al*., 2016). Temperature reconstructions for
the ANDRILL core were based on the TEX$_{86}$ index (McKay *et al*., 2012) and ODP 806 was analysed for Mg/Ca
of planktic foraminifera *Globigerinoides sacculifer* (Medina-Elizalde and Lea, 2010) renamed *Trilobatus
sacculifer* (Spezzaferri *et al*., 2015). ANDRILL sediment samples represent interglacial periods as organic
material at this location is not preserved during glacial intervals. These sediments are poorly constrained to
specific interglacial periods and are not assigned specific ages (McKay *et al*., 2012). Reported SST results exclude
sites ANDRILL and ODP 806 as HadISST, NZESM and UKESM cannot be produced for the ANDRILL site
(presently covered by the Ross Ice Shelf) and the NZESM high resolution model does not cover the region in
which ANDRILL and ODP806 are located, thus we cannot provide comparisons. Furthermore, they are not
alkenone-derived SST estimates.






**Table 1. Mid-Pliocene Warm Period (mPWP) site identification and location with associated surface water mass, sampling period and resolution (italicised in parenthesis), and source references for previously published data or age models used in association with new analyses.**

| Site | Latitude | Longitude | Surface Water Mass | Period (sampling) | Reference |
|---|---|---|---|---|---|
| ANDRILL1B | -77.889 | 167.089 | Antarctic Shelf Water | Interglacials during mPWP (3000-3300 ka) | McKay *et al.*, 2012 |
| DSDP594 | -45.524 | 174.948 | Subtropical Frontal Zone | 3000-3299 ka *(3 kyr)* | Caballero-Gill *et al.*, 2019; McClymont *et al.*, 2020 |
| ODP1172 | -43.960 | 149.928 | Subtropical Frontal Zone | 3000-3301 ka *(8 kyr)* | Age Model: Exon *et al.*, 2001; Data – this study |
| ODP1168 | -42.610 | 144.413 | Subtropical Frontal Zone | 3008-3290 ka *(7 kyr)* | Age model: Exon *et al.*, 2001; Data – this study |
| ODP1125 | -42.550 | -178.166 | Rekohu Eddy (extension of Tasman Front) | 3000-3299 ka *(2 kyr)* | Caballero-Gill *et al.*, 2019; McClymont *et al.*, 2020 |
| ODP1123 | -41.786 | -171.499 | Subtropical Water | 3004-3300 ka *(10 kyr)* | Age Model: Patterson *et al.*, 2018; Data – this study |
| DSDP593 | -40.508 | 167.675 | Subtropical Water (Tasman Sea) | 3025-3295 ka *(10 kyr)* | McClymont *et al.*, 2016; McClymont *et al.*, 2020 |
| DSDP590 | -31.167 | 163.3595 | Subtropical Water | 3017-3300 ka *(15 kyr)* | Age model: Karas *et al.*, 2011; Data – this study |
| ODP806 | 0.3185 | 159.361 | Western Pacific Warm Pool | 3000-3086 ka *(2 kyr)* | Medina-Elizalde and Lea, 2010 |

## 2.2 Biomarker ($U_{37}^{K'}$ and TEX$_{86}$) sea surface temperature reconstructions

Organic biomarkers preserved in marine sediments are important proxies for past water temperatures (e.g., de Bar *et al.*, 2019; Herbert *et al.*, 2010; Hollis *et al.*, 2019). The $U_{37}^{K'}$ index has been applied successfully to reconstruct SSTs in marine settings worldwide from low to high latitudes (e.g., Herbert, 2014). Although this proxy is calibrated to annual average SST using linear regressions based on sediment core top data between 60°N and 60°S (Müller *et al.*, 1998; Conte *et al.*, 2006; Rosell-Melé and Prahl, 2013), reconstructed SSTs can be biased towards higher temperatures due to peak alkenone production during the bloom period, which is commonly spring or early summer (Conte *et al.*, 2006; Prahl *et al.*, 2010). However, other studies used a combination of measurements and modelling to show that the maximum seasonality can be up to offset is ∼2.5°C at high latitudes (Conte *et al.*, 2006; Prahl *et al.*, 2010; Max *et al.*, 2020, McClymont *et al.*, 2020). To address the decreased response of $U_{37}^{K'}$ at high temperatures (>24°C), Tierney and Tingley (2018) developed a Bayesian B-spline regression model (BAYSPLINE). Previous studies, including some utilised here (e.g., McClymont *et al.*, 2020), applied the linear core top calibration of Müller *et al.* (1998). However because site DSDP 590 produces SST more than 24°C and there is little difference between the calibrations at mid-latitudes (maximum of 0.7°C), we have used the BAYSPLINE calibration and applied this to all sites (Appendix A). This results in slightly cooler temperatures (maximum <0.7°C; Table S1) ) but the difference remains within the calibration uncertainties (1.4°C below 24°C; Tierney and Tingley, 2018).

Additionally, for comparison with alkenone-based SST reconstructions, two sites (DSDP 590 and ODP 1172) were analysed for isoprenoid glycerol dialkyl glycerol tetraethers (isoGDGTs), which are produced by marine



Thaumarchaeota (Schouten *et al*., 2002; 2013) and used, to reconstruct TEX$_{86}$-derived SSTs. Because only a limited number of samples for two sites were analysed for TEX$_{86}$ (n=27) within the mPWP, the results are not used in analysis to determine reported means, but are discussed in Appendix A.


**2.3 Data analysis**

Data are summarised and visualised using R, an open access statistical software package (R Core Team, 2022). Probability distributions of the mPWP proxy SSTs, grouped by site, are displayed using 'vioplot' which graphically normalises the distribution for ease of comparison (Fig. 4). The plots often show a bimodal distribution curve which we infer to represent two normal distributions centred around mean interglacial and mean glacial

SSTs. Single mode distributions may reflect lower variability between glacial–interglacial conditions (e.g., low-latitude tropical sites), lower sample resolution that does not capture glacial–interglacial cyclicity, or sampling that favours either glacial or interglacial conditions (as is the case for ANDRILL, which is biased to interglacial ice retreat facies). Interglacials are typically identified through benthic δ$^{18}$O record cyclicity and tuning these

records to the global benthic δ$^{18}$O stack. However, glacial-interglacial cyclicity can be quite variable between different members in the stack during the mPWP (Lisiecki and Raymo, 2004), and this also occurs between records from the Southwest Pacific sites (e.g., McClymont *et al*., 2020). Furthermore, a number of these sites do not have δ$^{18}$O records and the SST records are not consistently cyclical or high-enough resolution to determine glacial and interglacials values. For that reason, we have employed a statistical package in R which identifies two modes that

are considered to represent average glacial and interglacial means, and thus, places more emphasis on values that record interglacial and deglacial transitions with less emphasis on glacial or interglacial extremes.

The temperature distributions for each site (excluding ANDRILL as interglacial values only) were assessed for bi-modal distribution to identify mean glacial and interglacial modes using the 'noramlmixEM2comp' function

in the R Package 'mixtools' (Benaglia *et al*., 2010). This employs an expectation-maximization (EM) algorithm to fit an equal two-component mixture model, assuming normal distributions. This is an automated process (samples are not identified as glacial or interglacial) assuming equal two-part mixture and normal distributions of these mixtures. While the accuracy of these results is dependent on the assumptions applied here - that the glacials and interglacials present a normal distribution and have an equal bi-modal component - it is a systematic approach

that applies statistical analysis to objectively identify the variance within the data and attribute that to glacial and interglacial conditions recorded in the data (Fig. B2). We acknowledge that this is an imperfect approach. However, we consider that this reduces bias introduced when visually selecting interglacial or glacial samples reliant on discrete values or temporal constraints (the latter are age model dependent). Secondly, this reduces emphasis on extreme warming during some interglacials of the mPWP and varying responses of the sites so we

can be more confident that the SSTs are reflective of the broader climate conditions of the mPWP. These interglacial modes are used for plotted and tabulated comparisons to the UKESM and NZESM projections presented in the results below. Uncertainty (1σ) associated with the $U_{37}^{K'}$-BAYSPLINE calibration is ±1.4°C below ~23.4° and non-linear above (Tierney and Tingley, 2018). Therefore, the higher SSTs of DSDP 590 have a higher uncertainty (average of ±2.4°C)(Table 2). The uncertainties for all proxy SSTs are taken as the mean of

all sites (±1.5°C) for absolute SST and when referenced to pre-industrial HadISST (Table 2).



## 3 Results

### 3.1 mid-Pliocene Warm Period Sea Surface Temperature signature

With respect to pre-industrial (HadISST), minimum paleo-SSTs for the mid-latitudes (45 to 30°S) are between 1.7–3.5°C, while mean site SSTs range from 0.8–6.6°C (average 3.4°C) with a likely (16$^{th}$ – 84$^{th}$ percentile range)

of 2–4.7°C (average for all sites) and maximum of 3.5 to 7.5°C (average 5.8°C) (Table 2). However, interglacial modal means range between 1.3–5.4°C (average 4.2°C) warmer than HadISST for the Southwest Pacific mid-latitude sites.

The sites presented in this study are sampled over glacial–interglacial cycles for which the total glacial–interglacial

amplitude of SSTs range from ~4.4–7.5°C (Fig. 4; Table 2) (excluding ANDRILL and ODP 806). Interglacial and glacial modal means determined by the bimodal statistical analysis (Section 2.3) are generally comparable to the 16$^{th}$ and 84$^{th}$ percentile (within ~1°C), highlighting that these modes are reflective of the likely range rather than accounting for extreme values representing the tails of the p-distribution used to estimate the total glacial-interglacial amplitude range, which has a mean of 6.1°C (Table 2). The difference between glacial and interglacial

modal means is approximately half the that of the total glacial-interglacial amplitude (~3°C; Table 2). The meridional gradients for mean glacials or interglacials do not differ significantly but do show a flattened gradient for interglacial modal means between site DSDP590 and ODP806 (30–0° S; Fig. 4b) due to the low SST distribution of site ODP806.

The sites warm (~0-20°C) from the pole to site ODP 1125 (north Chatham Rise), before a reduction in SSTs are seen at sites ODP 1123 (offshore Chatham Rise) and DSDP 593 (eastern Tasman Sea), then returning to high temperatures >25°C at sites north of 32°S (DSDP 590 and ODP 806) which show comparable peak temperatures (Fig. 4). While latitude is generally correlated with SST, surface water mass and regional currents alter this relationship. Site DSDP 594 south of the STF in surface Subantarctic Water is noticeably colder than sites situated

either within (ODP1172), or just north of (ODP1168, ODP 1125, ODP 1123), the STF.  However, current proximity to the STF doesn't  appear to be a main driver either.

DSDP 593 and DSDP 594 (north and south of the Subtropical Front) show the least warming above pre-industrial, but interglacial modal means still warm 1–2 °C. Sites that show significant interglacial modal mean warming

above the global mPWP average are offshore Tasmania (ODP 1168 and ODP 1172) and site ODP 1125 (northern Chatham Rise) which all display warming between 4.8–5.4°C, and DSDP 590 (north Tasman Sea) presents extreme warming of 6.7°C (Table 2).



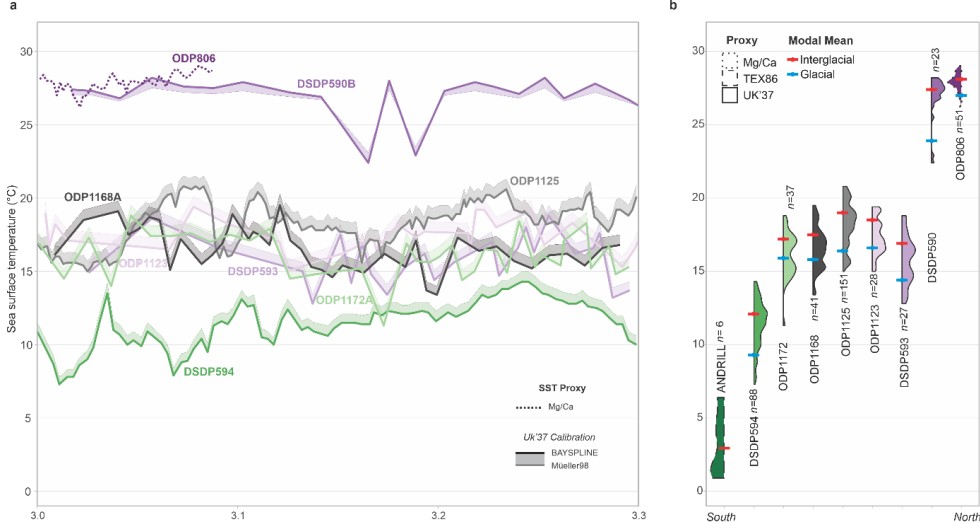

**Figure 4: Time series and SST distribution for mid-Pliocene Warm Period (3.3 – 3.0 Ma) records. a) $U_{37}^{K'}$ -SST calibrations of BAYSPLINE (solid; Tierney and Tingley, 2018) with the difference to Müller *et al*. (1998) shaded are plotted for all timeseries, with ODP 806 (Mg/Ca SST derived) and excluding ANDRILL (not plotted due to poor age control). b) Probability distributions ('violin plots') of the SST timeseries. Interglacial (red) and glacial (blue) modal means are also shown (Table 2; Fig. B1). Data for this plot are provided in Table S1 and Table S2.**

**Table 2. Statistical distribution of mid-Pliocene Warm Period Sea Surface Temperature (SST) anomalies relative to HadISST (1870-1879 AD) using $U_{37}^{K'}$ BAYSPLINE derived SSTs. Reported minimums, 16th percentile, mean (50th percentile), 84th percentile and maximum SST anomalies are shown with $U_{37}^{K'}$ BAYSPLINE reported 1σ uncertainty. Glacial and interglacial modal means are calculated as described in methods and the total range is calculated as the difference between maximum and minimum temperature.**

| Site | Min. (°C) | 16th (°C) | Mean (°C) | 84th (°C) | Max. (°C) | $U_{37}^{K'}$ ±σ (°C) | Glacial Modal-mean (°C) | Interglacial Modal-mean (°C) | Total range (°C) |
|---|---|---|---|---|---|---|---|---|---|
| DSDP594 | -3.5 | -0.9 | 0.8 | 2.2 | 3.5 | 1.4 | -1.5 | 1.3 | 7.0 |
| ODP1172 | -1.1 | 2.4 | 3.7 | 5.0 | 6.4 | 1.4 | 3.5 | 4.8 | 7.5 |
| ODP1168 | 0.8 | 2.6 | 4.0 | 5.3 | 6.9 | 1.4 | 3.2 | 4.9 | 6.1 |
| ODP1125 | 1.4 | 2.9 | 4.8 | 6.3 | 7.2 | 1.4 | 2.8 | 5.4 | 5.8 |
| ODP1123 | 0.7 | 2.0 | 2.9 | 4.5 | 5.1 | 1.4 | 2.3 | 4.2 | 4.4 |
| DSDP593 | -2.2 | -0.8 | 1.0 | 2.4 | 3.8 | 1.4 | -0.6 | 1.9 | 6.0 |
| DSDP590 | 1.7 | 5.8 | 6.6 | 7.2 | 7.5 | 2.4 | 3.2 | 6.7 | 5.8 |
| Mean | -0.3 | 2.0 | 3.4 | 4.7 | 5.8 | 1.5 | 1.8 | 4.2 | 6.1 |
| Variance | 5.2 | 6.7 | 5.8 | 5 | 4 | 1 | 5 | 5.4 | 3.1 |







### 3.2 Global Climate Models

#### 3.2.1 PlioMIP

Standardised boundary conditions used by all 16 models participating in PlioMIP, is termed the PlioCore experiment (Haywood *et al*., 2016; 2020), based on the latest PRISM4 climate reconstruction for MIS KM5c

(Dowsett *et al*., 2016). Data presented here is the multi-model mean of PlioCore (Haywood *et al*., 2020) with site specific SSTs extracted and referenced to the HadISST pre-industrial reanalysis (NCAR, 2022) for comparison to the mPWP proxy SSTs (Table 3; Fig. 5). Due to the poor spatial distribution of this study (although considerably higher than previous studies in the region), we are unable to sum the temperature distribution over latitudinal ranges of 30° for comparison to meridional gradients reported elsewhere (e.g., Haywood *et al*., 2020; McClymont

*et al*., 2020). However, we provide a comparison of mPWP site data to PlioCore latitudinal averages (1° resolution) between longitudes 140°E – 160°W, alongside site-specific SST from the PlioCore experiment (Haywood *et al*., 2020).

Specific site warming for PlioCore does not vary significantly from the meridional gradient except for ODP 1123

(Fig. 5b). Sites ODP 1123, DSDP 593 and DSDP 594 all present SST anomalies within 1°C for PlioCore and mPWP (Table 3). However, on average for the sites, PlioCore SST anomaly is 2.4°C, while mPWP is 4.2°C (Table 3).

**Table 3. Summary site mean Sea Surface Temperature (°C) for HadISST (1870-1879 AD), PlioCore (PlioMIP multi-model mean) and interglacial modal-mean mPWP $U_{37}^{K'}$ BAYSPLINE derived (this study), and SST Anomaly (relative to HadISST) for PlioCore and mPWP.**

|  | HadISST | PlioCore |  | mPWP |  |
| --- | --- | --- | --- | --- | --- |
|  | SST (°C) | SST (°C) | SST anomaly (°C) | SST (°C) | SST anomaly (°C) |
| DSDP594 | 10.8 | 12.7 | 1.9 | 12.1 | 1.3 |
| ODP1172 | 12.4 | 14.9 | 2.5 | 17.2 | 4.8 |
| ODP1168 | 12.6 | 16 | 3.4 | 17.5 | 4.9 |
| ODP1125 | 13.6 | 16.2 | 2.6 | 19 | 5.4 |
| ODP1123 | 14.3 | 17.6 | 3.3 | 18.5 | 4.2 |
| DSDP593 | 15 | 16.1 | 1.1 | 16.9 | 1.9 |
| DSDP590 | 20.7 | 22.6 | 1.9 | 27.4 | 6.7 |
| Mean | 14.2 | 16.6 | 2.4 | 18.4 | 4.2 |
| Variance | 9.9 | 9.9 | 2.3 | 15.3 | 5.4 |



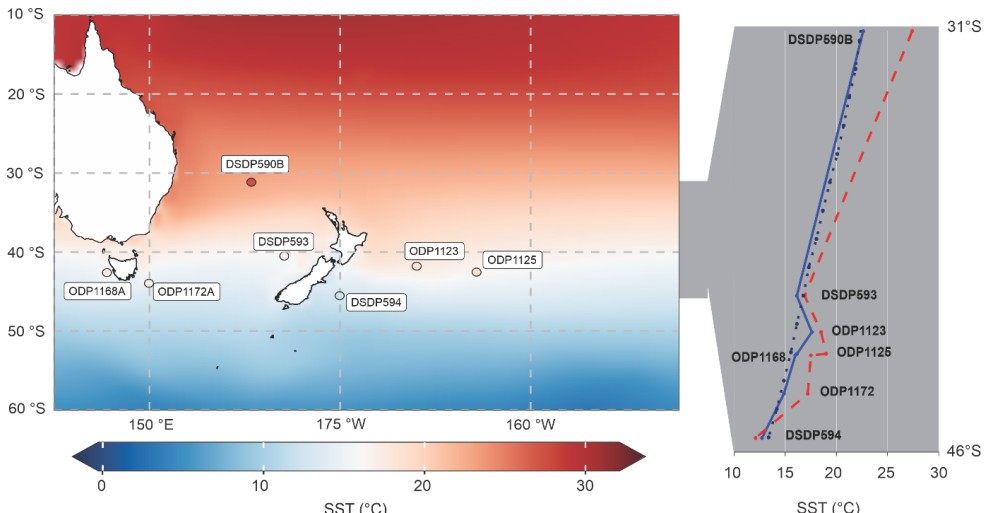

**Figure 5: Regional Sea Surface Temperature (SST) from PlioCore with mid-Pliocene Warm Period (mPWP) site mean**
**interglacial SST plotted using the same temperature scale. B) absolute SST between 31–46°S for PlioCore latitudinal**
**mean (blue dotted), PlioCore extracted sites (blue solid) and mPWP sites (red dashed).**

### 3.2.2    Future Earth System Model simulations

On average, NZESM simulations show higher warming than the coarser resolution UKESM in all scenarios
presented here (Table 4; Fig. 6). However, for the sites investigated, UKESM simulations shows more variability
between sites due to minimal warming at ODP594 and extreme warming at site ODP1172 (Table 4; Fig. 6). As
for proxy data in the previous section, statistical summaries refer to the mid–latitude sites (excluding ANDRILL
and ODP 806).

Projections for 2090–2099 AD for SSP1-2.6, SSP2-4.5 and SSP3-7.0 show a stable pattern of warming for both
models (Table 4; Fig. 6). However, warming at sites ODP1172, ODP 1168, ODP 1123 and ODP 1125 in UKESM
simulations increases above NZESM with higher emission scenarios, while DSDP 594 and 593 remain
significantly higher in NZESM simulations over UKESM (Table 4). NZESM and UKESM simulations for SSP3-
7.0 have similar mean warming (+4.5°C and +4.4°C respectively) to the mPWP (+4.2°C), with the means strongly
biased by differences in DSDP 594, 593 and 590 (Table 4).





**Table 4. Site annual mean Sea Surface Temperature anomalies (°C) for UKESM and NZESM with respect to HadISST**
**(1870-1879 AD) for SSP1-2.6, SSP2-4.5, SSP3-7.0 at 2095 AD (2090–2099 AD).**

| | 2090-2099 AD | | | | | | |
|---|---|---|---|---|---|---|---|
| | UKESM | | | NZESM | | | mPWP SST |
| Site | SSP 1 | SSP 2 | SSP 3 | SSP 1 | SSP 2 | SSP 3 | |
| | (°C) | (°C) | (°C) | (°C) | (°C) | (°C) | (°C) |
| DSDP594 | 0.1 | 1 | 2.3 | 2.3 | 3.3 | 4.9 | 1.3 |
| ODP1172 | 2.2 | 5.7 | 7.8 | 3.2 | 4.8 | 6.7 | 4.8 |
| ODP1168 | 0.6 | 3.1 | 5.4 | 1.4 | 2.7 | 3.9 | 4.9 |
| ODP1125 | 2.2 | 3.4 | 4.4 | 2.2 | 3.3 | 4.6 | 5.4 |
| ODP1123 | 2 | 3.4 | 4.3 | 1.1 | 2.2 | 3.5 | 4.2 |
| DSDP593 | 0.5 | 1.9 | 3.3 | 1.4 | 3 | 4 | 1.9 |
| DSDP590 | 1.3 | 2.4 | 3.4 | 1.8 | 2.8 | 3.9 | 6.7 |
| Mean | 1.3 | 3 | 4.4 | 1.9 | 3.2 | 4.5 | 4.2 |
| Variance | 2.1 | 4.7 | 5.5 | 2.1 | 2.6 | 3.2 | 3.2 |

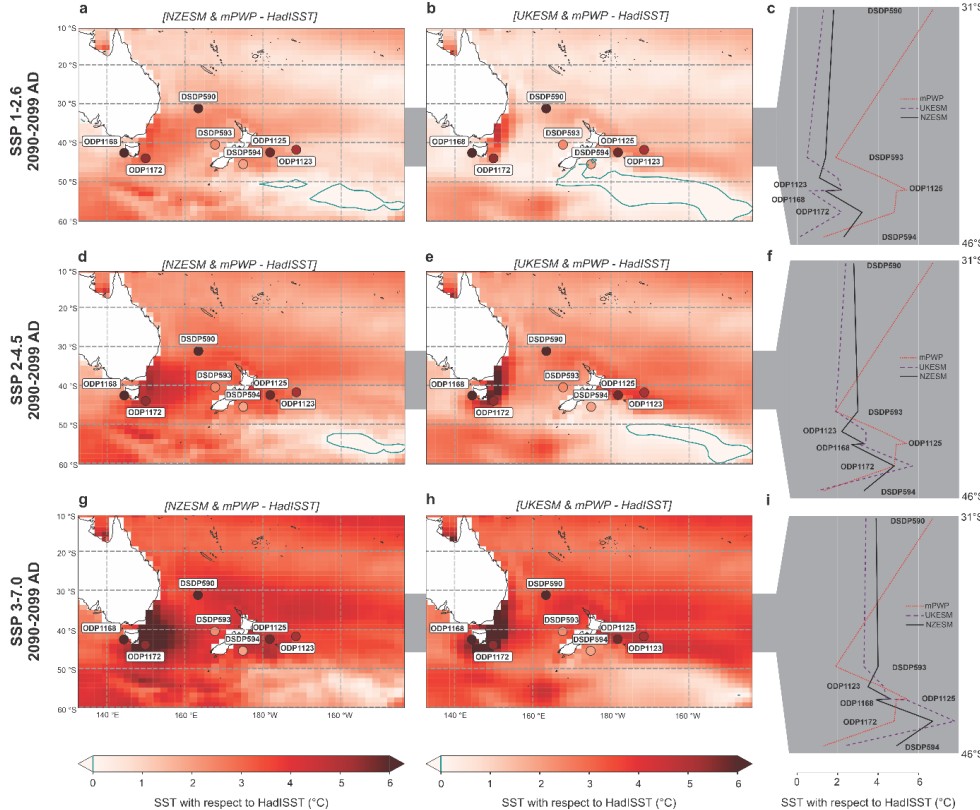

**Figure 6: Regional Sea Surface Temperature (SST) anomalies to HadISST (1870 – 1879 AD) for SSP1-2.6 (a–c), SSP2-**
**4.5 (d–f), SSP3-7.0 (g–i) in 2090–2099 AD compared to mid-Pliocene Warm Period (mPWP) site mean interglacial SST**
**anomalies (filled circles using same colour scale as map). Left panels are NZESM, middle panels are UKESM, and**
**right panels are site SST anomalies between 31–46°S for mPWP (red dotted), UKESM (purple dashed) and NZESM**
**(black solid).**



## 4    Discussion

### 4.1  Pliocene analogue

The mPWP encompasses several glacial cycles throughout the 300 kyr, with variable climate conditions (insolation, $CO_2$ and tropical SST; Fig. 2). Many studies have therefore focused on a single interglacial (MIS KM5c) as insolation values are near identical to today (Laskar *et al.*, 2004; Haywood *et al.*, 2020; McClymont *et al.*, 2020). However, this requires confidence in age models and ultimately tuning of records. The approach taken here, of assessing glacial and interglacial characteristics spanning the whole mPWP interval, aims to smooth glacial and interglacial extremes and represent the more "likely" climate conditions for equilibrium glacial and interglacial states across the region. Interglacial SST site modal means (this study) between 30° S and 45° S average at +4.2°C (Table 2) for global peak SST estimates of 2–3°C (Masson-Delmotte *et al.*, 2013). In comparison, for the same sites, PlioCore SSTs average 2.4°C (MIS KM5c ; Haywood *et al.*, 2020; Table 3), with global SSTs of 3.2°C (Haywood *et al.*, 2020; McClymont *et al.*, 2020).   Thus, this study demonstrates an amplified warming signal in the Southwest Pacific relative to the global mean temperature that is not recorded in the PlioCore simulations. Likewise, the mean of glacial modes is  +2°C with reference to HadISST (Table 2), where glacial conditions of the mPWP are often considered comparable to pre-industrial (Lisiecki and Raymo, 2005), which is however,  poorly studied. The comparable SSTs for sites with previously published values for MIS KM5c provides confidence in the approach of representing interglacial modal means used in this study and highlights the importance of regional variability in site selection to determine regional response (Fig. 7a). While we acknowledge the sites provided in this study are still spatially limited, they provide a significant increase to the resolution of sampling in this area for the mPWP.

Site DSDP 590 (northern Tasman Sea) presents the highest SSTs, which is currently north of the Tasman Front outlet of EAC. The location of the Tasman Front is controlled by the northern tip of New Zealand's' North Island, which was at a slightly lower latitude in the Pliocene (Strogen *et al.*, 2022) which may have allowed for a more northern Tasman Front, directing warmer waters across site DSDP 590. Alternatively, the warming at DSDP 590 may be explained by a broadening and invigoration of the Tasman Front, which may be at the expense of flow to the EAC-extension (Hill *et al.*, 2011). While a strengthening of the EAC is expected, the magnitude and distribution of that strengthening is argued (Hill *et al.*, 2011).  This circulation shift could also account for a lower degree of warming observed in the mPWP at site ODP 1172, situated in the southern extent of the EAC. Furthermore, redirected flow through the Tasman Front, which ultimately bathes the Chatham Rise, may account for the high degree of warming displayed by sites ODP 1123 and ODP 1125 (Table 2).

Furthermore, previous studies for the Last Interglacial (MIS 5e; 125 ka) suggest a warming of southern and eastern New Zealand (specifically based on data from site ODP 1123) may be a result of an increased and extended flow of the EAC becoming entrained in the Subtropical Front that would bathe the Chatham Rise sites (Fig 7a; Cortese *et al.*, 2013). The results presented here support a strengthening of the EAC and outlets relative to pre-industrial and modern, which is consistent with paleo studies for Late Pleistocene interglacials (Bostock *et al.*, 2015; Cortese *et al.*, 2013) and suggest these currents may have multiple ways of operating under warmer climates. Indeed, modern EAC transport and outlets are underestimated by most models (Chiswell *et al.*, 2015; Sen Gupta *et al.*, 2016, 2021).



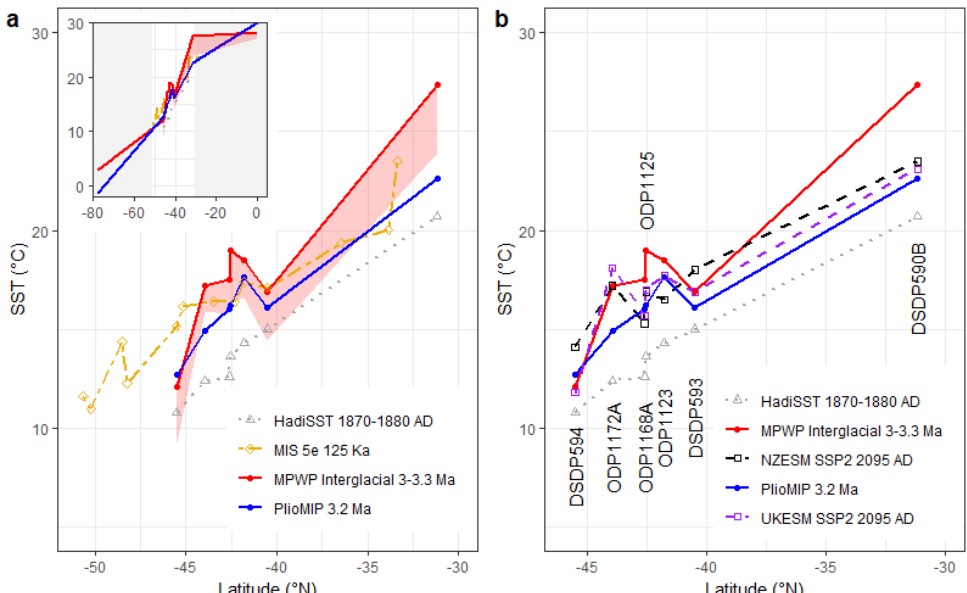

**Figure 7: Absolute Sea Surface Temperatures as a latitudinal transect of the Southwest Pacific with a) HadISST (NCAR, 2022), mid-Pliocene Warm Period interglacial (solid red) and glacial modal means (red ribbon), PlioMIP multi-model mean (solid blue) and Marine Isotope Stage (MIS) 5e (~125 ka) (dashed yellow; Cortese *et al*., 2013), and b) HadISST (dotted grey; NCAR, 2022), mid-Pliocene Warm Period interglacial modal mean (solid red), NZESM (dashed black) and UKESM (dashed purple) for SSP2-4.5 2090–2099 AD (Williams *et al*., 2016; Sellar *et al*., 2019).**

#### 4.1.1    Paleo – model comparison

Warming during the interglacial modal means of the mPWP can be simplified as >4°C above pre-industrial in five of the seven mid-latitude sites across the region, with two sites (DSDP 593 and DSDP 594) showing moderate warming (<2°C) (Table 2). This pattern is broadly reflected in both UKESM and NZESM projected scenarios explored here, with closer fit under middle of the road emission scenario (Fig. 6 & 7b; Table 4). NZESM and UKESM show a general trend (for the seven mid-latitude sites) of closer correlation to mPWP at lower temperature sites with increasing underestimation at sites with higher SSTs for all scenarios except for SSP3-7.0 (Fig. 6). These low temperature sites (DSDP 593 & DSDP 594) are also the two sites where UKESM provides systematically warmer values than NZESM (Fig. 6; Table 4).

The key differences between the UKESM and NZESM can be summarised by more distributed region-wide warming in the NZESM, with reduced warming along the EAC and offshore eastern New Zealand (Fig. 6). The pattern of NZESM SST field reflects local oceanographic grid refinement, which improves the fidelity of complex regional current transport and the representation of ocean fronts (Behrens *et al*., 2020). The concentrated regionally-limited warming of the UKESM, is less consistent with the mPWP signature of warming across the region (Fig. 6). Specifically, NZESM present lower SSTs for the EAC relative to the UKESM, which is more consistent with SSTs at site ODP 1172 during the mPWP (Fig. 6). This also corroborates the apparent intense warming observed at site DSDP 590 in the northern Tasman Sea during the mPWP, because of increased flow eastward to the Tasman Front at the expense of an invigorated EAC but may also reflect the paleogeographic



positioning of site DSDP 590 and the Tasman Front (Strogen *et al.*, 2022). Significantly, while the NZESM
produces a Subtropical Front further south than the UKESM, bathing DSDP 594 in warmer waters, this does not
extend to the eastern Chatham Rise sites, which is not consistent with mPWP observations (Fig. 6g–i). Lastly, we
note that warming at site ODP 1168 (southwest Tasmania) is comparable to site ODP 1172 (southeast Tasmania;
EAC) during the mPWP, which is inconsistent with both NZESM and UKESM (Fig. 6).


Modelled SSTs at the sites show increased variability under higher-emission scenarios (Fig. 6) more comparable
to the range and magnitude of mPWP observations, however, this is driven by closer values at DSDP 590 and
tends to overestimate SSTs at the other sites (Fig. 6; Table 4 & S5). Rather, for both UKESM and NZESM, SSP2-
4.5 for 2090–2099 AD show the least deviation to all sites reconstructed for the mPWP (Fig. 6d-f; & 7b). While

the magnitude of warming changes significantly with SSP projections, we consider both UKESM and NZESM
produce a pattern of warming consistent with site observations of the mPWP. We take this to suggest that
differences from paleogeography which potentially shifts ocean fronts and circulation (e.g., Haywood *et al.*, 2020)
during the mPWP have not resulted in significant changes to surface water mass distribution.

Considering regional SSTs in the meridional context, we have compared HadISST,  mPWP and PlioCore with
Last Interglacial MIS 5e (Fig. 7a) and ESM scenarios for SSP2-4.5 by 2090–2099 AD (Fig. 7b) . The glacial and
interglacial gradients of the mPWP are relatively consistent and show a much steeper gradient in comparison to
the interglacial MIS 5e (+1–2°C) and pre-industrial HadISST (Fig. 7a). While future ESM projections for SSP2-
4.5 2090-2099 AD (as the closest scenario to mPWP) shows much more comparable distribution with the mPWP

(Fig. 7b).

### 4.2  Global and regional warming

In comparing our interglacial modal mean mPWP SSTs, reconstructed using organic biomarker proxies, in
conjunction with data sampled for the same sites from transient ESM simulations to run to 2100 AD, we
acknowledge that the ESM values do not reflect the future equilibrium temperature responses for these Southwest

Pacific sites. However, in most cases, average regional temperatures at the sites studied are expected to increase
beyond 2100 AD as longer duration feedbacks in the Earth climate system play out. To consider the difference
between equilibrium and transient climates we discuss the variance of global and regional SSTs between paleo
and future scenarios.

Global SSTs during mPWP interglacials are ~3°C above pre-industrial (Masson-Delmotte *et al.*, 2013; Dowsett
*et al.*, 2016; McClymont *et al.*, 2020; Haywood *et al.*, 2020), comparable to expected warming (2.1–3.5°C) for
SSP2-4.5 by 2100 AD (IPCC, 2022). The pattern and magnitude of regional warming is similar between the
mPWP and ESM simulations under SSP2-4.5 (Table 4; Fig 6–7b), however the global warming generated by the
ESMs under SSP2-4.5 is 4°C. Thus, while these mPWP proxy SSTs present a higher degree of warming than

global, the same degree of warming from UKESM and NZESM requires 1°C higher global temperature increase.

The ECS of the UKESM (and NZESM) is 5.4°C (Sellar *et al.*, 2019; Senior *et al.*, 2020), which exceeds that of
estimates for the mPWP of 2.6–4.8°C (MIS KM5c; Haywood *et al.*, 2020), and far exceeds that of the accepted





range for the CMIP6 ensemble (2.5–4°C; IPCC, 2022). This is of importance because the Southern Ocean has
long been identified as having significant deviation from models to observations and it is uncertain whether high
ECS models (linked to shortwave cloud feedbacks; Zelinka *et al*., 2020) act to better estimate observations
(Schuddenboom and McDonald, 2021). Here, we show the high ECS simulations of NZESM and UKESM,
present a comparable warming signature seen during the mPWP in the Southwest Pacific, as opposed to the lower
ECS PlioCore simulations (Fig. 7b). These results demonstrate that while higher ECS models do produce more
extreme regional temperature response under transient climates and ~100 year-timescales, they require a higher
degree of global warming, suggesting longer-term feedbacks including ice dynamics may play a significant role
in accurately determining committed warming, particularly for this region in proximity to Antarctica and the
Southern Ocean. Furthermore, the use of lower ECS models (e.g. majority of the CMIP6 ensemble) for regional
downscaling in the Southwest Pacific may be underestimating the amplified warming signal we see in the mPWP
and ESM SSP2-4.5 scenarios.

## 5    Conclusions

The regional expression of warming differs from the global average on a variety of timescales and has significant
implications for the frequency and extent of climate induced hazards related to weather, sea-level rise and socio-
economic factors. Our mPWP proxy SST reconstructions for interglacial modal means show warming at sites
across the Southwest Pacific averaged at 4.2°C, that is 1-2°C above global warming (Masson-Delmotte *et al*.,
2013). This mPWP SST signature contains significant regional variability that is not seen in PlioCore multi-model
mean and exceeds the Southwest Pacific PlioCore average of 2.4°C (Haywood *et al*., 2020), but do replicate
warming at the three sites used in the PRISM climate reconstruction (Dowsett *et al*., 2016; McClymont *et al*.,
2020).

A flatter latitudinal SST gradient is seen for MIS5e (125ka) in comparison to the mPWP, however, warming
around Tasmania is consistent for the two periods and strongly suggests dynamic response of the East Australian
Current (EAC) under warmer climates. Indeed, modern observations suggest the invigoration of the Tasman Front
at the expense of the southward extent of the EAC could explain the intense warming at site DSDP 590 in the
northern Tasman Sea, as hypothesised by previous studies (Cortese *et al*., 2013; Bostock *et al*., 2015; Chiswell *et
al*., 2015; Sen Gupta *et al*., 2016; 2021).

The NZESM and UKESM show relatively consistent warming under low- and high-emission pathway
simulations, but the NZESM presents slightly warmer site averages in all scenarios. The most comparable
warming to mPWP by the ESMs is for 2090-2099 AD under the SSP2-4.5 scenario that is expected to reach 2.1-
3.5°C globally by 2100 AD (IPCC, 2022). However, the global warming for these ESMs under this pathway is
~4°C, which relates to the high ECS of the models. This suggests, that high ECS models better replicate the
regional warming signature in the Southwest Pacific, and that low ECS models in the CMIP6 ensemble may
underestimate warming in the Southwest Pacific. Ultimately, testing of longer-term scenarios using NZESM, to
accommodate for long feedbacks, for instance, potentially including a quantitative ice-sheet model (Smith *et al*.,
2021)**,** would provide insight into impacts of warming on ocean currents in the Southwest Pacific and determine
the effect of transient and equilibrium climate responses.



Paleoclimate reconstructions, such as those presented in this study, act as the only available evidence of equilibrium climate response to conditions predicted for the near future. While equilibrium climate states are not directly comparable to the transient future projections, expected sustained warming may result in comparable conditions.









**Appendix A**

Lipid biomarkers were analysed in the Organic Geochemistry Laboratory at GNS Science as reported in Naeher *et al*. (2012, 2014) and Ohkouchi *et al*. (2005) with modifications. In brief, freeze-dried and homogenized sediment samples (10–17 g) were extracted four times with dichloromethane (DCM)/ methanol (MeOH) (3:1, v:v) by ultrasonication for 20 min each time. Elemental sulfur was removed by activated copper. The total lipid extracts (TLEs) were divided into three fractions via liquid chromatography over silica columns using *n*-hexane (F1), *n*-hexane/DCM (1:2, v:v; F2) and DCM/MeOH (1:1, v:v; F3), respectively.

The F2 fractions containing alkenones were analysed using gas chromatography mass spectrometry (GC-MS) on an Agilent 7890A GC System, equipped with an Agilent J&W HP-1ms capillary column [60 m × 0.32 mm inner diameter (i.d.) × 0.25 µm film thickness (f.t.)], and connected through a splitter to an Agilent 5975C inert MSD mass spectrometer and flame ionisation detector (FID). The oven was heated from 70°C to 280°C at 20°C min$^{-1}$, then at 4°C min$^{-1}$ to 320°C and held isothermal for 20 minutes with a total run time of 40.5 minutes. Helium was used as carrier gas with a constant flow of 1.0 mL min$^{-1}$. Samples (1 µL) were injected splitless at an inlet temperature of 320°C. The MS was operated in electron impact ionisation mode at 70 eV using a source temperature of 230°C. For alkenone identification, the MS was operated in simultaneous full scan and single ion monitoring (SIM) mode at *m/z* 55, 58, 97, 109.1, 526.5, 528.5 and 530.5. Alkenones were quantified using FID. Glycerol dialkyl glycerol tetraethers (GDGTs), present in the F3 fractions, were dissolved in *n*-hexane/isopropanol (99:1, v:v) and filtered with 0.45 µm PTFE filters prior to liquid chromatography mass spectrometry (LC-MS) analysis at the University of Hokkaido, Japan. GDGTs were analysed on an Agilent 1260 HPLC system coupled to an Agilent 6130 Series quadrupole MS. Separation was achieved using a Prevail Cyano column (2.1 × 150 mm, 3 µm; Grace Discovery Science, USA) maintained at 30°C following the method of Hopmans *et al*. (2000) and Schouten *et al*. (2007). Conditions were: flow rate 0.2 ml/min, isocratic with 99% *n*-hexane and 1% isopropanol for the first 5 min followed by a linear gradient to 1.8% isopropanol over 45 min. Ionization was achieved using atmospheric pressure, positive ion chemical ionization. The spectrometer was run in selected ion monitoring mode (*m/z* 743.8, 1018, 1020, 1022, 1032, 1034, 1036, 1046, 1048, 1050, 1292.3, 1296.3, 1298.3, 1300.3, and 1302.3). Compounds were identified by comparing mass spectra and retention times with those in the literature (Hopmans *et al*., 2000).

The $U_{37}^{K'}$ index is defined based on the relative abundance of the $C_{37:2}$ and $C_{37:3}$ alkenones according to Prahl and Wakeham (1987) as follows:

$$U_{37}^{K'} = [C_{37:2}]/([C_{37:2}] + [C_{37:3}]) \tag{1}$$

We used the calibration of Müller *et al*. (1998) and BAYSPLINE (Tierney and Tingley, 2018) to reconstruct SSTs from the $U_{37}^{K'}$ index.

The TEX$_{86}$ index is based on the relative distribution of isoprenoidal glycerol dialkyl glycerol tetraethers (isoGDGTs) in marine sediments, originally defined by Schouten *et al*. (2002):



$$TEX_{86} = \frac{[GDGT-2]+[GDGT-3]+[GDGT-4']}{[GDGT-1]+\{GDGT-2\}+\{GDGT-3\}+\{GDGT-4'\}} \qquad (2)$$

where GDGT-1, GDGT-2 and GDGT-3 are characterized by one, two and three cyclopentane moieties and cren′ is the regioisomer of crenarchaeol. This index derived from core top samples was calibrated to SSTs using linear regressions as proposed by Schouten *et al* (2002) and Kim *et al.* (2008).

To test the reliability of reconstructed SSTs and to increase confidence in the choice of the applied calibrations,
we have compared $U_{37}^{K'}$ and TEX$_{86}$ SST at two sites. While $U_{37}^{K'}$ SSTs using the BAYSPLINE (Tierney and Tingley, 2018) does yield slightly cooler temperatures (up to 0.7 °C) at higher-latitude sites than the calibration of Müller *et al.* (1998), the TEX$_{86}$ SSTs differ by +6.4 to -16.9°C dependent on the calibration used (Figs. A1, A2). This proxy may be compromised at sites with high soil organic matter inputs (Hopmans *et al.*, 2004) and high contributions of sedimentary GDGTs (Pancost *et al.*, 2001; Zhang *et al.*, 2011) which is considered negligible
in open-marine environments. Other non-temperature controls such as oxygen concentrations, growth phases, nutrient cycling may be introduced in upwelling zones but are not able to be addressed here due to limited understanding of these effects (Elling *et al.*, 2014; Qin *et al.*, 2015; Hollis *et al.*, 2019). Non-linear calibrations such as the $TEX_{86}^{H}$ index (Kim *et al.*, 2010) were developed to extend the calibrated SST range of the previous calibrations, however this may underestimate SSTs in ancient greenhouse climates (Tierney and Tingley, 2015;
O'Brien *et al.*, 2017, Hollis *et al.*, 2019) and a non-linear relationship contradicts available experimental evidence suggesting a linear relationship with SST (Pitcher *et al.*, 2010; Schouten *et al.*, 2013; Elling *et al.*, 2014). Therefore, a Bayesian approach (BAYSPAR; Tierney and Tingley, 2015) was developed to consider spatially varying uncertainty derived from modern SST distribution is widely used. Additionally, a new machine-learning approach (OPTiMAL: Optimised Palaeothermometry from Tetraethers via MAchine Learning) aims to address
uncertainty in the method application to paleo SST and determine SST beyond the modern range (>30°C) (Dunkley Jones *et al.*, 2020). In comparing the two independent biomarker proxies of derived SST using $U_{37}^{K'}$ - BAYSPLINE with TEX$_{86}$ calibrations of Schouten *et al.* (2002), Kim *et al*, (2010), Tierney and Tingley (2015), and Dunkley Jones *et al.* (2020), we find all calibrations are comparable (~±5°C) but the BAYSPAR approach of Tierney and Tingley (2015) displays the closest values to $U_{37}^{K'}$-BAYSPLINE (Fig. A1 & A2; Table A1). Notably,
the calibrations of Schouten *et al.* (2002), Kim *et al.* (2010) and BAYSPAR (Tierney and Tingley, 2015) show less scatter at higher temperatures (>25°C; Fig. A1), while the OPTIMAL calibration (Dunkley Jones *et al.*, 2020) presents offsets of up to -15°C (Fig. A2) in comparison to $U_{37}^{K'}$-BAYSPLINE.

The TEX$_{86}$ calibration of Tierney and Tingley (2015) (BAYSPAR) shows the closest values to the $U_{37}^{K'}$ - SST BAYSPLINE and lowest scatter (Figs. A1, A2; Table A1), and therefore are selected for display in Fig. 4. In
contrast, the calibration of Schouten *et al.* (2002) shows larger scatter in reconstructed SSTs than BAYSPAR (Figs. A1, A2). The calibration of Kim *et al.* (2010) yields similar SST estimates as Schouten *et al.* (2002) and BAYSPAR, but seems to overestimate SSTs at lower temperatures. In contrast, OPTIMAL (Dunkley-Jones *et al.*, 2020) appears to underestimate SSTs at higher temperatures. Importantly, the general agreement in SST reconstruction from two independent biomarker provides higher confidence in the results.






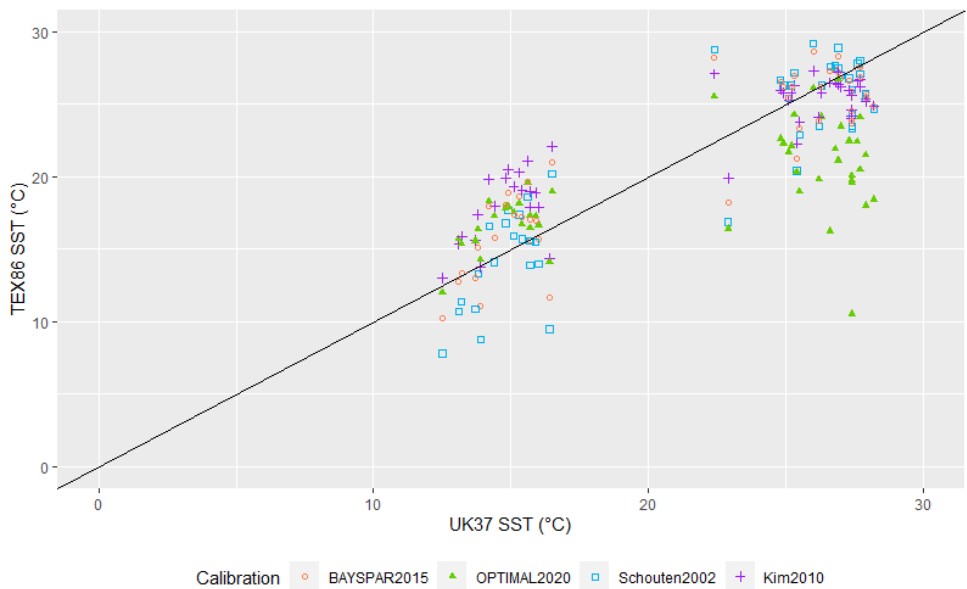

**Figure A1: Comparison between $U^{K'}_{37}$ derived SST using BAYSPLINE with TEX$_{86}$ Index SST calibrations of Schouten *et al.* (2002), Kim *et al*, (2010), Dunkley Jones *et al*. (2020) and Tierney and Tingley (2015).**

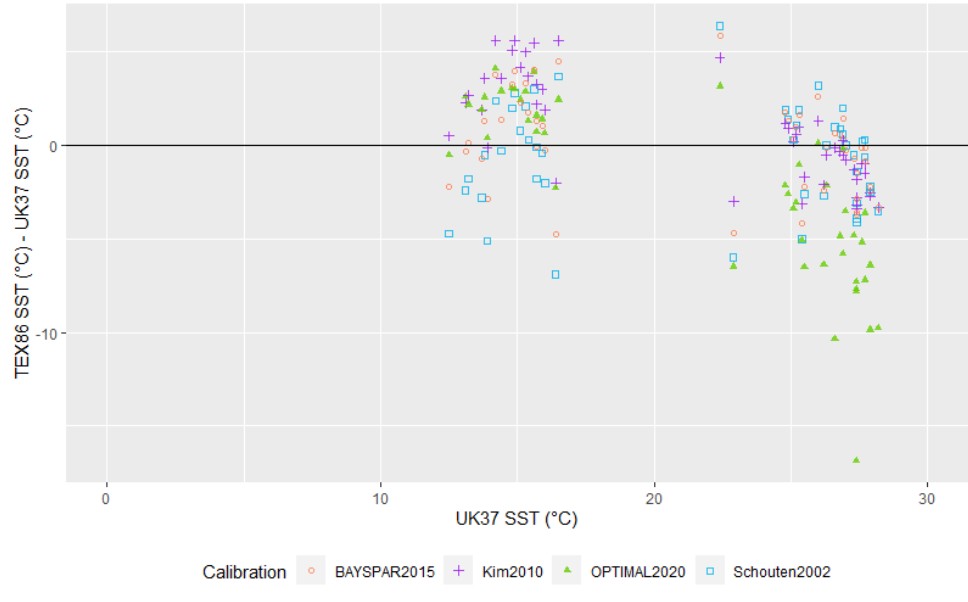


**Figure A2: Comparison between $U^{K'}_{37}$ derived SST using BAYSPLINE with TEX$_{86}$ Index SST calibrations of Schouten *et al.* (2002), Kim *et al*, (2010), OPTIMAL (Dunkley Jones *et al*., 2020) and BAYSPAR (Tierney and Tingley, 2015).**

**Table A1: Comparison between $U^{K'}_{37}$ derived SST using BAYSPLINE with TEX$_{86}$ Index SST calibrations of Schouten**
***et al.* (2002), Kim *et al*, (2010), OPTIMAL (Dunkley Jones *et al*., 2020) and BAYSPAR (Tierney and Tingley, 2015).**



| TEX$_{86}$ calibrations | Average Difference (°C) of TEX$_{86}$ SST –relative to BAYSPLINE SST reconstructions |
|---|---|
| BAYSPAR 2015 | 0.1 |
| Kim2010 | 0.8 |
| OPTIMAL2020 | -2.3 |
| Schouten2002 | -0.6 |

**Appendix B**

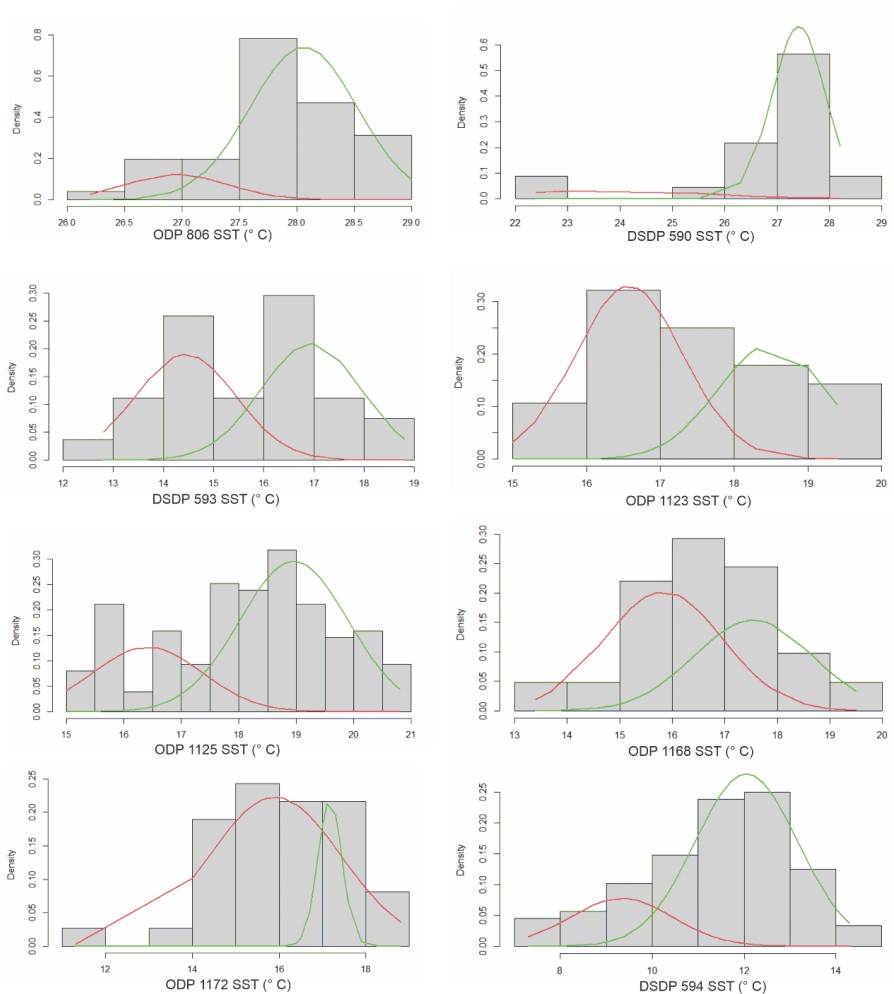

**Figure B2. Bimodal analysis for each site after Benaglia *et al*., (2010), excluding ANDRILL as it only represents interglacial conditions, displaying density curves with calculated bimodal distributions interpreted as glacial distributions (red) and interglacial (green). Code is available.**



**Code and Data availability**

Data tables and supplementary tables: DOI 10.5281/zenodo.7109199

Script and necessary data files: https://github.com/GRG-GNS/Pliocene-SST-Southwest-Pacific

Table S1. All site sea surface temperature (SST; °C) data used in results with index and calibrations of Müller98 (Müller *et al*., 1998) and BAYSPLINE (Tierney and Tingley, 2018). The proxy type and references are also provided.

Table S2. Site sample data for analyses undertaken this study, including all and TEX$_{86}$ index calculations and
calibrations. References for calibrations are contained within column headers.

Table S3. Seasonal and annual mean sea surface temperature (SST; °C) model outputs of HadISST (NCAR, 2022), UKESM (Sellar *et al*., 2019), NZESM (Williams *et al*., 2016) at the seven Southwest Pacific sites (DSDP 594, ODP 1172, ODP 1168, ODP 1125, ODP 1123, DSDP 593, DSDP 590) for SSP2 2040 AD (2036-2045 AD), and SSP1, 2, and 3 2095 AD (2090-2099 AD). Including UKESM and NZESM with respect to
HadISST.

Table S4. Site sea surface temperature (SST; °C) annual means and seasonal range for UKESM and NZESM SSP2-4.5 2036-2045 AD, with MPWP interglacial modal means and total glacial range (maximum to minimum SST).

Table S5. Compiled sea surface temperature (SST; °C) interglacial means for MIS 5e (125kyr; Cortese *et al*.,
2013) and mPWP (3.3-3.0 Ma) and model annual means for HadISST (1870-1879 AD), and SSP2-4.5 2090-2099 AD for UKESM, NZESM.

**Sample availability**

Samples were obtained from the International Ocean Discovery Program, Texas A&M University.

**Author Contribution**

GRG, JHTW and SN designed the project. SN, OS and MY measured and analysed the data. JHTW and AMH provided climate model simulations. GRG prepared the manuscript with the contribution of all authors.

**Competing interests**

The authors declare that they have no conflict of interest.

**Acknowledgements**

We thank the International Ocean Discovery Program (IODP), which provided the samples, and the Australia-New Zealand IODP Consortium (ANZIC), which provided Legacy Analytical Funding for this study. ANZIC is supported by the Australian Government through the Australian Research Council's LIEF funding scheme [LE160100067] and the ANZIC of universities and government agencies. We acknowledge support from GNS Science and the New Zealand Ministry of Business, Innovation and Employment through the Global Change
Through Time research program and Organic Geochemistry Laboratory (contract C05X1702). JW obtained funding and support through the Ministry of Business Innovation and Employment Deep South National Science Challenge project number C01X1412. The development of the UKESM, was supported by the Met Office Hadley Centre Climate Programme funded by BEIS and Defra (GA01101) and by the Natural Environment Research



Council (NERC) national capability grant for the UK Earth System Modelling project, grant number
NE/N017951/1. JW wishes to acknowledge the use of New Zealand eScience Infrastructure (NeSI) high
performance computing facilities, consulting support and training services as part of this research. New Zealand's
national facilities are provided by NeSI and funded jointly by NeSI's collaborator institutions and through the
Ministry of Business, Innovation & Employment's Research Infrastructure programme, www.nesi.org.nz. OS
acknowledges the support by Japan Society of Promotion of Science funded by Ministry of Education, Culture,
Sports, Science and Technology, Japan (KAKENHI 26287129 and 17H06318). We also wish to acknowledge the
invaluable sea surface temperature compilations and assessment by PlioVar (Pliocene Climate Variability), a Past
Global Changes (PAGES) working group and the contributing authors.

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
