# Peer review of "Amplified surface warming in the Southwest Pacific during the mid-Pliocene (3.3–3.0 Ma) and future implications"

_EGUsphere, 2023_

## Author Response (AR1)

We thank Reviewer 1 for their kind review, succinct description of the manuscript and helpful edits to improve the readability and quality of the manuscript.

Below, we list each of the comments provided by the reviewer in plain text and our responses to them in red text, with new text added to the manuscript in red italics. When noting our changes, we refer to line numbers in the new version of the manuscript.

Comments:

Line 103-104. The sentence is misleading. Anyway, the warm mPWP is an equilibrated climate. Whether the future climate (unequilibrated) can reach the mPWP condition remains unknown. We agree with the reviewer and have edited the sentence to reference temperatures specifically by remove implied assumption that climate conditions will be similar. Line 104 "*Global temperatures (+2-3°C) last experienced during the mPWP (3.3-3.0 Ma) may be reached by 2100 AD …*"

Figure 3. It is difficult to read Figure 3. Please consider replotting the figure with lines and shaded areas. We agree with the reviewer and have replotted Figure 3 (Line 210) as a simple line plot for NZESM (dashed) and UKESM (solid). As they are overlapping we are unable to show the range (with transparency) in runs associated with each model for each SSP scenario.

Figure 4. It is difficult to distinguish proxy in plot b. We agree with the reviewer and have adjusted Figure 4b (Line 370) where proxy Mg/Ca has a light grey dashed outline and proxy TEX has a solid outline, while the majority (proxy UK37) has no outline.

Figure 5. The caption should be revised; consider using "Ensemble mean regional SST from core PlioMIP experiments". We have edited as per your suggestion, and further edited the use of PlioCore in the caption and manuscript to simplify references for the reader. In response to Reviewer 2 we have also added a range of values for PlioMIP to represent the range in ensemble members.

In Table 4, please add the simulated global mean. It will be helpful to show if the simulated regional warming exceeds the global mean. We have edited as per your suggestion to include the global mean of each model for each SSP.

Figure 6. The gradient plots look misleading. The mismatch between the simulated and the reconstructed gradient is mainly due to the reconstructed strong warming in DSDP590. Therefore, it is better to add some discussion about the gradient when the warming at DSDP590 is discussed. We agree with the reviewer that the anomalous warming at DSDP590 skews the visual representation of latitudinal gradients, and while we have referenced this intense warming, we have not emphasised the interpretation of this in the context of assessing gradients. Thus, we have added the text Line 425 "*The intense warming recorded at site DSDP 590 during the mPWP is particularly visible in latitudinal gradient comparisons (Fig. 6c,f,i) and highlights the importance of comparing site specific data.*"

Line 456, "recorded" is not a good word to describe simulations. We have changed to "displayed" Line 457

Line 571-575, it is better to remove this from the conclusion. In this study, the authors do not provide solid evidence to show the change in EAC. Even though the models underestimate the

warming at DSDP590, they should show some signals for the EAC change. However, the authors do not investigate that. We agree with the reviewer that this is an over statement from the result presented here, so this has been removed from the conclusions. Statements that the results presented here are consistent with modern observations and other studies has been retained in the discussion.

Reviewer 2:

We thank Reviewer 2 for their attention to detail and considerate, helpful comments on the challenges this study addresses. The reviewers comments have added rigor to the manuscript.

Below, we list each of the comments provided by the reviewer in bold text and our responses to them in rplain text, with edited text added to the manuscript in italics. When noting our changes, we refer to line numbers in the new version of the manuscript.

To summarise broadly, introductory statements by the reviewer touches on the issue of equilibrium and transient climate comparison, the range of model responses especially with regard to equilibrium climate sensitivity and reference sea surface temperatures used. We consider these comments to be valid concerns  and to this end we have included two more models (CESM2 ECS – 5.2K [UKESM ECS 5.4K]and INM ECS – 1.3 K) for comparison in Appendix C (described further in technical comment response below). The reference sea surface temperatures (using individual model pre-industrial runs for reference) is more problematic as an NZESM pre-industrial run has not been undertaken which is why we have compared all data to HadISST as the most referenced historical reanalysis. To the challenge of comparing transient and equilibrium (paleo) climates, we feel this is well understood by the community and we can not solve this issue here. However, in our treatment of the paleo data – taking the full glacial to interglacial range and interglacial modal means, we do not feel we are overstating the warming experienced at these sites in the mPWP (i.e., by using peak SST values which are considerably warmer). Furthermore, the conclusions of the study are based on a comparison of regional warming to global warming specific to the time periods (~2100 AD and ~3 Ma). However, the concept of the paper does intend to encourage the reader to think of continued climate warming beyond a 2100 AD timeframe, but we were careful not to conclude that the Pliocene climate state is what the future holds, only that past warming patterns should be considered as a real result of limited global warming in which to interpret current warming.

**L86: I find this sentence either wrong or vague. Many estimates show Last Glacial Maximum climate sensitivity to be weaker than modern/warmer paleoclimates due to state-dependency on climate feedbacks (see PALAEOSENS, 2012, for instance). Yes, warm-based estimates are usually more consistent with high ECS; however, you cannot summarize that the entire paleoclimate record relates to high ECS**. We acknowledge this statement is fundamentally incorrect by encompassing all paleo climates. We have edited the text to specify warmer-than-present with reference to the summary by the IPCC WGI. Line 86: "Higher ECS is more consistent with estimates of *warmer-than-present* paleo climate sensitivity (*Forster et al., 2021).*" This will be discussed in more detail on various ECS estimates in 1.1 Paleoclimate analogues.

**L119: Why cite Haywood et al. (2016) here, and not Haywood et al. (2011)?** This was in error, citation changed to 2011.

**L120: Connected to previous comment; I do not fully understand this sentence or even where those values come from. As I said in Technical comment, I cannot find your reference of Haywood et al. (2012), but I will assume you refer to Haywood et al. (2013). In that paper, the only number I could find for fast-feedback ECS would exceed 3 K. Also, if you wish to be thorough with ECS, you cannot ignore Hargreaves and Annan (2016), which effectively combined PRISM data and PlioMIP1 models and came up with 1.9 - 3.7 K, or more recently Renoult et al. (2020) with 0.5 - 5.0 K. I assume the numbers you provide are in range of median values, where in that case both median estimates of Hargreaves and Annan (2016) and Renoult et al. (2020) fall in the range of 2 - 3 K, and in that case you should actually say that you are showing median estimates. From a first read, I immediately thought you were presenting confidence intervals, and that your old ECS estimates are very well-constrained and low, in particular compared to PlioMIP2 estimates.** We acknowledge this information was not representative of the methods available and have added further description of ECS estimates for clarity, based on the IPCC report.

*Line 120-129: "The Pliocene Modelling Intercomparison Project (PlioMIP) presents a multi-model ensemble with various ECS run for mPWP conditions (Haywood et al., 2011; 2020). The recent IPCC summary of ECS (Forster et al., 2020) does not include model-based estimates, but methods for mPWP paleoclimate ECS based on emergent constraints (Hargreaves and Annan, 2016; Renoult et al., 2020) utilising PlioMIP (Haywood et al., 2011; 2020) and proxy temperature and $CO_2$ reconstructions (Martinez-Boti et al., 2015; Sherwood et al., 2020), range from median values of 2.5–3.7°C. Marine Isotope Stage (MIS) KM5c (3.2 Ma) interglacial became a focus for reconstructing warming within mPWP as insolation values and the orbital configuration were most similar to the Holocene interglacial (Haywood et al., 2020; McClymont et al., 2020). While, based on less data points, this approach has better agreement between models and observations and revealed a higher ECS of 2.6–4.8°C for conditions of MIS KM5c from the PlioMIP Phase 2 ensemble (PlioMIP2; Haywood et al., 2020).*

**L192: I find weird and essentially a circular argument to say that the ECS of UKESM is higher than the IPCC range, since the high end of the IPCC range is itself calculated/constrained by models such as UKESM. This is similar as saying "UKESM lies on the high end of the IPCC range which itself is based on UKESM high ECS". I think this requires either a rephrasing, or maybe getting rid of the IPCC reference.** We have reworded to clarify that the UKESM (as a CMIP member) has a high ECS in reference to the total CMIP ensemble, and removed the IPCC reference.

*Line 192 "The UKESM (as a CMIP ensemble member) and NZESM have an ECS of 5.4°C (Sellar et al., 2019) which is higher than the likely range (high confidence) for ECS as 2.5–4°C (Zelinka et al., 2020).*

**L201 - 205: I do not fully get this part. The way I understand it is that you calculated temperature anomalies of models between a perturbed (non-pre-industrial state) model run and an observational dataset, which implies that inherent biases in the models are kept in your perturbed model temperature. Most likely your pre-industrial climate is not at zero top-of-atmosphere radiative imbalance, which you could have corrected by comparing the perturbed to the modelled pre-industrial state. I understand you used HadISST for proxy data, and I completely agree on not using a model control here though. But here the issue is that you will obviously have consequences on your model temperatures which are not accounted for and which come from model physics and simplifications, or even numerical approximations made by using different clusters.** Unfortunately, the NZSEM does not have a

long-period pre-industrial run for comparison. We initially started with reference to the UKESM pre-industrial but realised the model bias from NZESM to UKESM was exacerbated by this process. We acknowledge that using model pre-industrial reference states would be best practice, but as it is unavailable, we have used HadISST as the most referenced historical SST reanalysis.

**L338 - 340: Is there a better way of introducing those results? In one sentence we are given 4 ranges of temperature and 2 mean values for the same sites, and I quickly lost the thread.** We have split this sentence into two for clarity and have been more consistent with referring to the figures.

*LINE 337-341: "With respect to pre-industrial (HadISST), mean site SSTs for the mid-latitudes (45 to 30°S) range from 0.8–6.6°C with a likely (16th – 84th percentile range) of 2–4.7°C (3.4°C average for all sites). Minimum SST ranges for the sites are -3.5–1.7°C (-0.3°C average for all sites) and maximum SSTs range from of 3.5 to 7.5°C (5.8°C average of all sites) (Table 2). Interglacial modal means, used in this study as moderate warm conditions, range between 1.3–5.4°C (average 4.2°C) warmer than HadISST for the Southwest Pacific mid-latitude sites."*

**L455: Here you report a PlioCore SST anomaly (3.2C), which is calculated as the anomaly between Pliocene ("perturbed") state and control. However, you compare it to the PlioCore SST average, which I guess is calculated from the difference between model and proxy (if I read Table 3 of the reference). However, I know that McClymont et al. (2020) have temperature anomalies of around 3.2C, so I would guess it is just a weird referencing. I would simply get rid of the Haywood et al. (2020) reference in that line.** We have edited this comparison and referencing more clearly, using McClymont's reference of SSTs as global proxy reconstructions. PlioMIP results for the sites are referenced to the global multi-modal mean, including standard deviation and percentiles of ranges, as per next comment. Uncertainties for the McClymont study and this study are standard alkenone calibration uncertainties which have been stated previously and do not add further meaning here.

*Line 453-456 " Interglacial SST site modal means (this study) between 30° S and 45° S average at +4.2°C (Table 2) for global SST estimates of 3.2-3.4°C (McClymont et al., 2020). In comparison, for the same sites, PlioMIP SSTs average 2.4±2.1°C (Haywood et al., 2020; Table 3), with global multi-mode median of 3.0°C (10th-90th percentiles: 2.1-4.8°C) (Haywood et al., 2020)."*

**L454 - 455: Overall I have a bigger issue because there is no uncertainty given on those values. You say the PlioCore simulations do not record the amplified warming signal in Southwest Pacific, but what is the full range of the simulated temperature there?** We agree the range in PlioMIP simulations are meaningful and have included ranges in comment above and plotted minimum, maximum and ±1σ of the PlioMIP simulations to the Fig.5, with the mean to maximum also plotted for the SSTs presented in this study.

[Figure]

**L521 - 523: I get the point but it is also speculative. You could have many other things which influence the SST at that site in a model and coincidentally make it close to the reconstructed SST at that site (e.g. clouds), and then you could conclude that changes due to paleogeography actually influence surface water masses distribution. A way of approaching this problem would be to run PlioCore with UKESM / NZESM, although I understand this would be an entire new study.**

We acknowledge this is speculative and have removed the statement from Line 522-524.

**L547: What is "accepted" range? This is the likely range. The very likely would put max ECS at 5, and then I would not consider 5.4 to far exceed it. Also, again, this argument is circular. The range of the IPCC you are providing is dependant of UKESM. It works better when comparing to mPWP, since there UKESM was not used**.

Text altered to 'likely range' not 'accepted' and it is previously specified that the UKESM is in the CMIP ensemble. It is well-known the UKESM is a high ECS model comparative to the majority of ESMs and to provide that context, we refer to the range of CMIP6 models.

 *Line546-548"The ECS of the UKESM (and NZESM) is 5.4°C (Sellar et al., 2019; Senior et al., 2020), which exceeds that of estimates for the mPWP of 2.6–4.8°C (MIS KM5c; Haywood et al., 2020), and far exceeds that of the likely range (2.5–4°C) for the CMIP6 ensemble (Forster et al., 2021).*

**L552 - 554: In Fig.7b you show the average PlioMIP state. Do you also find a comparable warming signature in those regions for the other high ECS models (of e.g. the PlioMIP2 ensemble)? CESM2 has a comparable, even higher ECS than UKESM / NZESM (5.6 K, see Zhu et al. (2022)), or even COSMOS could show something similar. If it is not the case, what you are seeing might be only due to UKESM (and the similarities it has with NZESM), and not a systematic behaviour of high ECS models. It seems your argument mainly relies on the ECS of the models, so you would not need a high resolution one to test it on multiple models.**

We thank the reviewer for the suggestion and have included model results for CESM2 (ECS 5.2 – 5.6K; Danabasoglu et al., 2020; Zelinka et al., 2020) and INM (ECS 1.8K; Volodin et al., 2018) extracted for the sites and referenced to HadISST for comparison in Appendix C. These models support the results and show that high ECS and low ECS do indeed behave different in this region. Because this does not alter the results, and including the figure would require major revision of the manuscript to include descriptions of the models and further discussion that we don't feel would significantly improve the study, we have added this figure as an Appendix for interest and refer to this figure in the final statement of the discussion.

*Line 557-559: "Furthermore, the use of lower ECS models (e.g. majority of the CMIP6 ensemble) for regional downscaling in the Southwest Pacific may be underestimating the amplified warming signal we see in the mPWP and ESM SSP2-4.5 scenarios (Appendix C)."*

[Figure]

*Figure C1. Extended version of Figure 6 to include a second high ECS model (CESM2: ECS 5.2-5.6°C; Danabasoglu et al., 2020) and the lowest ECS model in CMIP6 (INM: ECS 1.8°C; Volodin et al., 2018). Regional Sea Surface Temperature (SST) anomalies to HadISST (1870 – 1879 AD) for SSP1-2.6 (a–c), SSP2-4.5 (d–f), SSP3-7.0 (g–i) in 2090–2099 AD compared to mid-Pliocene Warm Period (mPWP) site mean interglacial SST anomalies (filled circles using same colour scale as map). Panels (a-c) are NZESM, panels (d-f) are UKESM, panels (g-i) are CESM2, panels (j-l) are INM and the far-right panels (m-o) are site SST anomalies between 31–46°S for mPWP (red dotted), UKESM (purple dashed) and NZESM (black solid), CESM2 (dark green solid) and INM (light green dashed). The INM low ECS model shows a significantly different pattern of warming to the high ECS models of NZESM, UKESM and CESM2.*

Technical comments:

**L75: cryosphere** corrected

**In bibliography: Where is Haywood et al., 2012? I could not find it.** Error in referencing, changed to Haywood et al., 2011 and added to references

**L285: Weird sentence ("seasonality can be up to offset...")** Edited to 'seasonality variations can be up to ~2.5°C….'

**L302 and elsewhere in this section: Details about using R are unnecessary, since you could have used any other language combined with statistical modules to perform those analyses**

**(Python could have done that)**. Specific reference to R has been removed, with the exception of the specific package used in R to determine mixed modal means.

**L350: "half the that", remove the.** Removed 'the'

**L359: The STF? I forgot what it was and I could not search for it, then realised it is in your Fig.1... I would write it once more in the text.** All reference to STF has been removed and is now in text written in full to avoid confusion.

**L360: remove the "," before "the STF".** As above

**L363: Here you talk about Subtropical Front but do not use the acronym STF though...** As above

**L511: Similar as before, here STF is not used.** As above

**Fig.7 : I would not plot lines, as there is such a limited amount of data, in particular it seems weird to connect with a straight line point at around -40 and -30**. We acknowledge that connecting lines between point data across sparse latitudinal data may be considered misleading, however we attempted several plotting techniques and found line plots were the best visual representation with care taken for colour blind readers hence the mix of bold and dashed lines. We hope that the spatial maps help to draw the reader's attention to the real difference in latitude.

---

## Author Response (AR2)

To the Editor,

Thank you for your comments and close review of the manuscript. I failed to see your last report and apologise for not addressing those comments in the response to reviewers.

We have adjusted the manuscript text as below, but I wanted to take the opportunity to add further explanation here. As stated in our response to reviewers' comment, the lack of a NZESM pre-industrial run was the key reason for using HadISST as a reference. This decision was further validated as we dealt with numerous model and data sources with various means of reporting. Specifically – the PlioMIP dataset is a model ensemble mean, and the LIG proxy data was referenced to another source. However, in acknowledgment of that potential bias, we plotted the absolute temperatures in Fig 5 and 7 when comparing all data and from all data sources.

**L. 122-126:** This sentence is very convoluted and hard to understand. Please amend it.

*Edited from: "The recent IPCC summary of ECS (Forster et al., 2021) does not include model-based estimates, but methods for mPWP paleoclimate ECS based on emergent constraints (Hargreaves and Annan, 2016; Renoult et al., 2020) utilising PlioMIP (Haywood et al., 2020) and proxy temperature and $CO_2$ reconstructions (Martinez-Boti et al., 2015; Sherwood et al., 2020), range from median values of 2.5–3.7°C. "*

*To: "The recent IPCC summary of ECS ranges from median values of 2.5–3.7°C (Forster et al., 2021). This ECS summary does not include model-based estimates, but does include emergent constraints (Hargreaves and Annan, 2016; Renoult et al., 2020) utilising PlioMIP (Haywood et al., 2020) and proxy temperature and $CO_2$ reconstructions (Martinez-Boti et al., 2015; Sherwood et al., 2020). "*

**L. 196: Remove "as" in front of "a CMIP"** *Removed*
**L. 197: remove "at" in front of higher.** *Removed*
**L. 209-210: The last part of the sentence does not make much sense.** *This is edited as following to also clarify the use of HadiSST further and lack of an NZESM pre-indsutrial run.*

*"Best practise of model assessment is to present anomalies with reference to pre-industrial runs from the same model. As a pre-industrial run is unavailable for the NZESM, we have used the single reference of HadISST for all model and proxy anomaly assessment. HadISST was selected as it is the most complete reanalysis product nearest to pre-industrial conditions."*

---

## Author Response (AR3)

Dear Laurie,

Again, thank you for your attention to detail. I have edited the manuscript as follows:

Line 339 (in clean manuscript): "With respect to pre-industrial (HadISST),  mean site SST anomalies for the mid-latitudes (45 to 30°S) range from 0.8–6.6°C with a likely (16th – 84th percentile range) of 2–4.7°C (3.4°C average for all sites). Minimum SST anomalies for the sites range from -3.5–1.7°C (-0.3°C average for all sites) and maximum SST anomalies range from  3.5 to 7.5°C (5.8°C average of all sites) (Table 2). Interglacial modal mean anomalies, used in this study as moderate warm conditions, range between 1.3–5.4°C (average 4.2°C) warmer than HadISST for the Southwest Pacific mid-latitude sites."

A last thank you to the reviewers, you Laurie as the handling editor and the journal for accepting the manuscript. The process has been very responsive, clear and easy to follow the progress.

Kind regards,

Georgia Grant